# Ectopic CH60 mediates HAPLN1-induced cell survival signaling in multiple myeloma

Debayan De Bakshi[1,2,3], Yu-Chia Chen[2,3], Shelly M Wuerzberger-Davis[2,3], Min Ma[4], Bayley J Waters[5], Lingjun Li[4,6], Aussie Suzuki[2,3,7], Shigeki Miyamoto[2,3,7]

**Multiple myeloma (MM), the second most common hematological malignancy, is generally considered incurable because of the development of drug resistance. We previously reported that hyaluronan and proteoglycan link protein 1 (HAPLN1) produced by stromal cells induces activation of NF-κB, a tumor-supportive transcription factor, and promotes drug resistance in MM cells. However, the identity of the cell surface receptor that detects HAPLN1 and thereby engenders pro-tumorigenic signaling in MM cells remains unknown. Here, we performed an unbiased cell surface biotinylation assay and identified chaperonin 60 (CH60) as the direct binding partner of HAPLN1 on MM cells. Cell surface CH60 specifically interacted with TLR4 to evoke HAPLN1-induced NF-κB signaling, transcription of anti-apoptotic genes, and drug resistance in MM cells. Collectively, our findings identify a cell surface CH60-TLR4 complex as a HAPLN1 receptor and a potential molecular target to overcome drug resistance in MM cells.**

## Introduction

Multiple myeloma (MM), a cancer of the antibody-producing plasma B cells, is the second most common hematological malignancy and is considered incurable according to current clinical standards (Ravi et al, 2018). The International Agency for Research on Cancer estimated in 2018 that global MM incidence and mortality amounted to 160,000 cases and 106,000 patients, respectively (Ferlay et al, 2019). Despite the introduction of combination therapy, in which more traditional drugs (e.g., dexamethasone) are used in conjunction with newer drugs (Kumar & Rajkumar, 2018), such as proteasome inhibitors (e.g., bortezomib), immunomodulatory therapies (e.g., lenalidomide), and monoclonal antibodies (e.g., daratumumab), development of drug resistance is still a major cause of patient mortality (van de Donk et al, 2021).

MM cells are home to the BM, and the BM tumor microenvironment (TME) provides a crucial niche of cell-extrinsic factors that enables MM cells to proliferate, survive, migrate, and manifest drug resistance. MM cells have been shown to respond to a gamut of soluble growth factors and pro-survival cytokines present in the MM TME, such as IL-6, IGF-1, TNF-α, SDF-1, and BAFF, that promote drug resistance and disease progression (Hideshima et al, 2007). Cross-talk between MM cells and cellular agents of the MM TME, such as osteoclasts (Abe et al, 2004), macrophages (Zhang et al, 2021), and bone marrow stromal cells (BMSCs) (Nefedova et al, 2003; de Jong et al, 2021), has also been shown to protect MM cells from the chemotherapy-induced cytotoxicity. Furthermore, interactions of structural ECM proteins of the MM TME with their cognate adhesion receptors on the MM cell surface, such as fibronectin with α5β1 integrin (Hazlehurst et al, 2000) and laminin-1 with 67LR (Vande Broek et al, 2001), have been reported to induce drug resistance in MM cells.

The ECM contributes to cancer progression by providing both mechanical and chemical cues (Winkler et al, 2020). Recent work has broadened this paradigm by investigating the role of "matrikines," peptide fragments produced by partial proteolysis of structural ECM proteins. These ECM-derived peptides act as signaling elements that bind to cell surface receptors and exert biological activities that differ from the structural role of the full-length counterparts (Maquart et al, 2004). Matrikines are reported to elicit a wide array of tumor-permissive functions in solid tumors (Marinkovich, 2007; Grahovac et al, 2013). For example, the proteolytic processing of laminin-332 releases distinct domains, which are capable of engaging a wide variety of receptors on squamous cell carcinomas, such as α6β4 integrin, syndecan-1, and EGFR, to promote cancer cell survival, invasion, and migration (Marinkovich, 2007). However, the role of matrikines is not limited to tumor promotion, as they also exhibit tumor-inhibitory activities (Hamano et al, 2003; Lambert et al, 2018). As examples, collagen-derived matrikines tumstatin and tetrastatin act through the αvβ3 integrin receptor to inhibit angiogenesis-dependent lung carcinoma growth and melanoma progression, respectively (Hamano et al, 2003;

[1]Cellular and Molecular Biology Graduate Program, University of Wisconsin, Madison, WI, USA    [2]McArdle Laboratory of Cancer Research, University of Wisconsin, Madison, WI, USA    [3]Department of Oncology, University of Wisconsin, Madison, WI, USA    [4]School of Pharmacy, University of Wisconsin, Madison, WI, USA    [5]Department of Cell and Regenerative Biology, University of Wisconsin, Madison, WI, USA    [6]Department of Chemistry, University of Wisconsin, Madison, WI, USA    [7]University of Wisconsin Carbone Cancer Center, University of Wisconsin, Madison, WI, USA

Correspondence: smiyamot@wisc.edu

Lambert et al, 2018). Although the role of ECM-derived matrikines in the modulation of pathogenesis and progression of solid tumor types is well documented, their role in hematological malignancies, and MM in particular, remains obscure.

Previously, our laboratory reported that a matrikine derived from hyaluronan and proteoglycan link protein 1 (HAPLN1), also known as cartilage link protein 1 or link protein (Spicer et al, 2003), can induce drug resistance in MM (Huynh et al, 2018). HAPLN1 is a glycoprotein that conventionally functions as an essential part of the cartilage ECM by providing strength and flexibility (Hardingham, 1979). HAPLN1 enacts this physiological endeavor through three distinct domains: an N-terminal immunoglobulin-like (IG) domain that binds to the proteoglycan monomer aggrecan, and two C-terminal proteoglycan tandem repeat (PTR1/2) domains, which tether this protein complex to hyaluronic acid (HA) polymers. Beyond its presence in cartilage, HAPLN1 is also ubiquitously expressed in many other tissues within both the developing vertebrate embryo and the adult (Binette et al, 1994). Homozygous HAPLN1-mutant mice exhibit a severe phenotype because of defects in cartilage development and delayed bone formation, resulting in short limbs, craniofacial anomalies, and early postnatal death because of respiratory failure (Watanabe & Yamada, 1999).

In the context of MM disease, we reported that HAPLN1 was secreted as a soluble factor by MM patient–derived BMSCs, but not by MM cells themselves (Huynh et al, 2018). Soluble HAPLN1 was detected in MM patient BM plasma in both larger forms containing all three domains, and smaller forms containing the PTR1/2 domains but lacking the IG domain. The smaller forms were shown to be enriched in relapsed/refractory MM patients with progressive disease. We further showed in MM cancer cells that the PTR1 domain of HAPLN1 (HAPLN1-PTR1), but not full-length protein, activated the transcription factor NF-κB, a mediator of inflammation and cell survival that has been implicated in cancer progression. An HA-binding mutant of HAPLN1-PTR1 continued to elicit an NF-κB response at a comparable level to WT HAPLN1-PTR1, suggesting that the classical ECM function of HAPLN1-PTR1 to bind HA was unnecessary for its signaling ability. We also showed that HAPLN1-PTR1 was sufficient to induce drug-resistant survival in MM cells against the drug bortezomib, an indispensable component of MM first-line therapy. Finally, we reported that full-length HAPLN1 is processed into the HAPLN1 matrikine by matrix metalloproteinase 2, also produced by MM patient–derived BMSCs (Mark et al, 2022). We concurrently showed that MM cells up-regulated HAPLN1 synthesis and matrix metalloproteinase 2 activity in BMSCs to promote the generation of this matrikine and cause drug resistance. Collectively, these studies revealed that a novel HAPLN1 matrikine produced in the MM TME activates pathobiological NF-κB and induces drug resistance in MM cells. However, the molecular mechanism by which MM cells detect HAPLN1-PTR1 to engage downstream protumorigenic NF-κB signaling and drug resistance remains unknown.

In the current study, we sought to identify the putative HAPLN1-PTR1 receptor present on MM cell surface membrane. We identified aberrantly localized chaperonin 60 (CH60), also known as heat shock protein 60, and classically considered a chaperone protein present in the mitochondria (Nisemblat et al, 2015), as the direct binding partner of HAPLN1-PTR1 on the MM cell surface. We employed super-resolution imaging to show that CH60–TLR4 complexes preexist on MM cells and genetic, pharmacological, and biochemical analyses provide evidence that CH60 acts via TLR4 to induce NF-κB signaling, up-regulation of anti-apoptotic genes, and drug resistance in MM cells. Our study reveals that HAPLN1 matrikine engages a novel CH60-TLR4 signaling complex on MM cells to protect them from drug-induced cytotoxicity. This previously undescribed HAPLN1-PTR1 receptor complex on the MM cells may provide a potential therapeutic target to modulate the protumorigenic effect of HAPLN1 without altering its vital role in cartilage biogenesis and maintenance of ECM structure.

# Results

### Generation of recombinant HAPLN1-PTR1 peptide with improved NF-κB signaling dynamics

Our earlier studies showed that in MM cells, the PTR1 domain of HAPLN1 (HAPLN1-PTR1), when purified as a GST fusion protein, elicited an NF-κB response and drug resistance (Huynh et al, 2018). We noted that most of GST-fused HAPLN1-PTR1 (GST-PTR1) partitioned into insoluble inclusion bodies, unavailable for use. Furthermore, the soluble protein fraction tended to precipitate out of solution during removal of contaminants and impurities. Thus, GST-PTR1 limited the ability to biochemically identify a putative cell surface receptor on MM cells. To alleviate this technical hurdle, we first aimed to create high-quality recombinant HAPLN1-PTR1 that can be isolated at higher concentrations in a stable and soluble form possessing greater biological activity. We leveraged the use of a maltose-binding protein (MBP)–fusion tag to generate an MBP-fused HAPLN1-PTR1 peptide (MBP-PTR1) in an *Escherichia coli* system; MBP is known to promote stable solubility in the fused peptide due its intrinsic chaperone-like activity (Kapust & Waugh, 1999), attract bacterial chaperones such as GroEL (Douette et al, 2005), and translocate the fused peptide to a cellular location, which has less protease content and a more favorable redox environment for protein folding (Nallamsetty et al, 2005).

After expression in a Rosetta strain of *E. coli*, and partial purification by binding to and eluting from an amylose column, soluble MBP-PTR1 fractions were subjected to endotoxin removal via poly-L-lysine resin. The resulting protein preparations were then subjected to successive gel filtration and anion-exchange chromatography, yielding monomeric, soluble, and near-pure MBP-PTR1 protein with almost no detectable degradation products (Fig 1A). Experiments measuring relative changes in NF-κB activity are performed using electrophoretic mobility shift assay (EMSA) analysis with Oct-1 binding as a loading control for each sample, unless otherwise specified. The first EMSA images shown (Fig 1B) display the entire gels, showing the shifted transcription factors relative to the free probe line. All EMSAs are performed in the presence of excess probe to ensure that availability of probe is not a limiting factor in quantifying the relative activation of the transcription factors being measured. Subsequent EMSA images are shown with only the shifted transcription factor areas being depicted. On treating RPMI8226 human MM cells with equal total protein amounts of MBP-PTR1 fractions after each purification step, the specific activity of NF-κB response normalized to Oct-1 binding

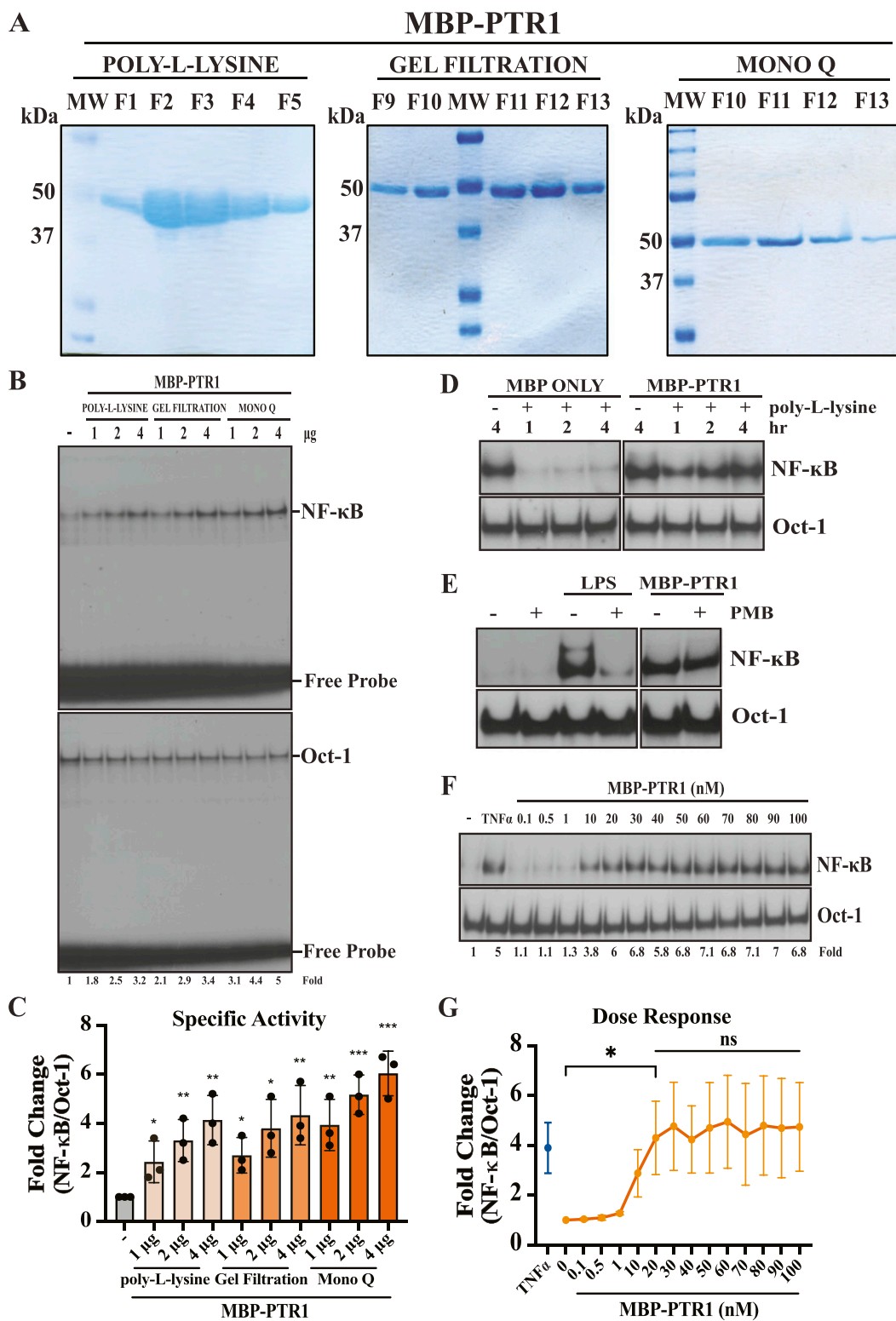

**Figure 1. Optimized generation of recombinant MBP-PTR1 improves PTR1-mediated NF-κB activity in MM cells.**
**(A)** Coomassie blue staining of MBP-PTR1 fusion peptide–eluted fractions after sequential steps of poly-L-lysine (to remove endotoxin), gel filtration, and MonoQ anion-exchange enrichment is shown as a ~50-kD protein band. **(B)** EMSA analysis of RPMI8226 cells incubated with indicated amount of total protein present in eluted MBP-PTR1 fractions from each subsequent purification step for 2 h. **(C)** Quantification of fold change of NF-κB activation relative to unstimulated control from three independent experiments included as mean ± SD. *$P < 0.05$; **$P < 0.01$; and ***$P < 0.001$ ($t$ test). **(D)** EMSA analysis of RPMI8226 cells treated for 2 h with 100 nM of MBP-only control or MBP-PTR1 before and after poly-L-lysine step. **(E)** EMSA analysis of RPMI8226 cells treated with 100 nM of MBP-PTR1 after poly-L-lysine step for 2 h in the

increased (Fig 1B and C). This indicates that the monomeric MBP-PTR1 moiety is concordantly enriched and responsible for eliciting the NF-κB response. In addition, after MBP-only peptide was subjected to endotoxin removal via poly-L-lysine resin, the ability of the protein preparation to activate NF-κB was abrogated (Fig 1D, left panel). However, similarly processed MBP-PTR1 continued to strongly activate NF-κB in RPMI8226 cells (Fig 1D, right panel). We also tested the impact of polymyxin B, a cationic polypeptide that is known to inhibit the critical "lipid A" portion of endotoxin (Morrison & Jacobs, 1976), and found that it prevented endotoxin (LPS)-induced NF-κB activation, but failed to cause any reductions in activation induced by MBP-PTR1 (Fig 1E). To further rule out the possibility that NF-κB activation was due to the presence of contaminating endotoxin, we measured the amounts of endotoxin in MBP-PTR1 preparations (Fig S1A). The levels of contaminating endotoxin were found to be < 0.15 ng/ml, and the final endotoxin concentration that RPMI8226 cells were treated with was at least three orders of magnitude lower because of dilution into cell culture media. The final endotoxin concentration after dilution is ~4 orders of magnitude below the threshold required for detectable NF-κB activation in RPMI8226 cells as we previously reported (Huynh et al, 2018). This confirms that HAPLN1-PTR1 is responsible for the NF-κB activity stimulated by MBP-PTR1 preparations, and not endotoxin contamination nor the MBP-fusion tag.

In our earlier publication (Huynh et al, 2018), we had shown that 100 nM of GST-PTR1 elicits saturable NF-κB activity in RPMI8226 cells. In contrast, MBP-PTR1 elicited saturable NF-κB activity at 20 nM in the same cell line, a fivefold gain in functional activity (Fig 1F and G). Consistent with eliciting NF-κB activity as measured by EMSA, MBP-PTR1 also concordantly induced phosphorylation of the inhibitor of nuclear factor-κB (IκB) kinase (IKK) complex and degradation of IκBα in RPMI8226 cells (Fig S1B). Simultaneously, pretreatment of RPMI8226 cells with IKK16, a chemical inhibitor of the IκB kinase (IKK) complex (Waelchli et al, 2006), was shown to block MBP-PTR1–mediated stimulation of the NF-κB signaling cascade (Fig S1B). We also observed that MBP-PTR1 was able to induce transcription of NF-κB–regulated target genes (Fig S1C), NF-κB inhibitor alpha (NFKBIA/IκBα) (Sun et al, 1993), and NF-κB subunit 1 (NFKB1/p105) (Ten et al, 1992). These results collectively indicate that MBP-PTR1 possesses the ability to evoke canonical elements of the NF-κB signaling cascade. Among human MM cell lines tested, RPMI8226 cells showed the strongest NF-κB activation response, similar to an optimal dose of a well-established NF-κB activator, TNF-α (Fig 1F and G). MM.1S and MM.1R cell lines showed lower levels of activation compared with RPMI8226 cells, and U266 cell line showed little to no detectable activation (Fig S1D–F). Where indicated, EMSA was performed in the presence of RelB antibody to super-shift basal RelB complexes (which are not activated by HAPLN1-PTR1) to enable quantification of canonical NF-κB complexes activated by HAPLN1-PTR1, as reported previously (Huynh et al, 2018).

We also showed that GST-PTR1 promotes bortezomib resistance in MM cell lines in an in vitro toxicity assay (Huynh et al, 2018), but we could not test whether GST-PTR1 also induced bortezomib resistance in vivo because of its poor fidelity. Because MBP-PTR1 is well behaved (i.e., it can be isolated at high concentrations in a highly pure form, remains soluble while subject to clean-up, and does not precipitate over time after freeze–thaw cycles), we next tested its ability to induce bortezomib resistance in an MM tumor xenograft model in mice. In the absence of bortezomib therapy, MBP-PTR1 provided no significant growth advantage to MM.1S and RPMI8226 tumors compared with MBP-only control (Fig 2A and B). However, when the tumor-bearing mice were challenged with bortezomib therapy, MBP-PTR1 promoted significantly greater growth of MM tumors relative to MBP-only control (Fig 2C and D). MM.1S tumors at 14 and 20 d of bortezomib therapy appeared grossly distinct between MBP-only and MBP-PTR1 groups (Fig 2E and F). Upon histological examination of the tumors isolated at 14 d, the MBP-only group displayed severe necrotic features, whereas the MBP-PTR1 group had apparently healthy MM tumor cells (Fig 2G). These data collectively demonstrate that MBP-PTR1 is superior to GST-PTR1 in its ability to induce higher NF-κB signaling in vitro and cause bortezomib resistance in vivo in human MM cells. We therefore proceeded to identify a putative HAPLN1-PTR1 receptor on MM cells using MBP-PTR1.

## HAPLN1-PTR1 specifically interacts with CH60 at the plasma membrane of MM cells

Among the human MM cell lines tested, RPMI8226 cells demonstrated the most robust NF-κB signaling upon MBP-PTR1 treatment (Figs 1F and S1D–F). Thus, we employed this cell line and leveraged the Sulfo-SBED biotin label system (Alley et al, 2000) to identify a putative receptor for HAPLN1-PTR1. MBP-PTR1 and MBP-only were labeled with the Sulfo-SBED biotin tag, and their biotinylation was confirmed by streptavidin-HRP Western blotting (Fig 3A, top and middle panel). Upon confirming that biotin-labeled MBP-PTR1 was still able to cause NF-κB signaling in MM cells (Fig 3A, bottom panels), intact RPMI8226 cells were incubated for 15 min at 4°C with biotin-labeled MBP-PTR1 to promote bait–prey engagement at the plasma membrane, while blocking potential bait internalization into intracellular compartments. In parallel, the same experiment was performed with the addition of a 10-fold stoichiometric excess of unlabeled bait to test for specificity. Cell surface biotinylation was then performed by UV irradiation as described in the Materials and Methods section, followed by subcellular fractionation of RPMI8226 cells and Western blot analysis of the protein fractions with streptavidin HRP. This analysis revealed that MBP-PTR1 specifically interacted with a ~55-kD prey localized at the plasma membrane, which was absent in the cytosolic fraction (Fig 3B). Subsequent solubilization of membrane proteins followed by gel filtration further detected the ~55-kD prey with a

presence and absence of 10 µg/ml polymyxin B, along with a positive control LPS stimulation at 50 ng/ml for 2 h **(F)** EMSA analysis of RPMI8226 cells showing a dose–response of MBP-PTR1 treated for 2 h. RPMI8226 cells were incubated with 10 ng/ml TNF-α for 15 min as a positive control. **(G)** Quantification of fold NF-κB activation relative to unstimulated control from three independent experiments included. ns, not significant; *P < 0.05 (t test). Fold change of NF-κB DNA binding as measured by phosphor-image quantification, corrected for Oct-1 DNA binding control, and normalized to unstimulated control is labeled below the gel.

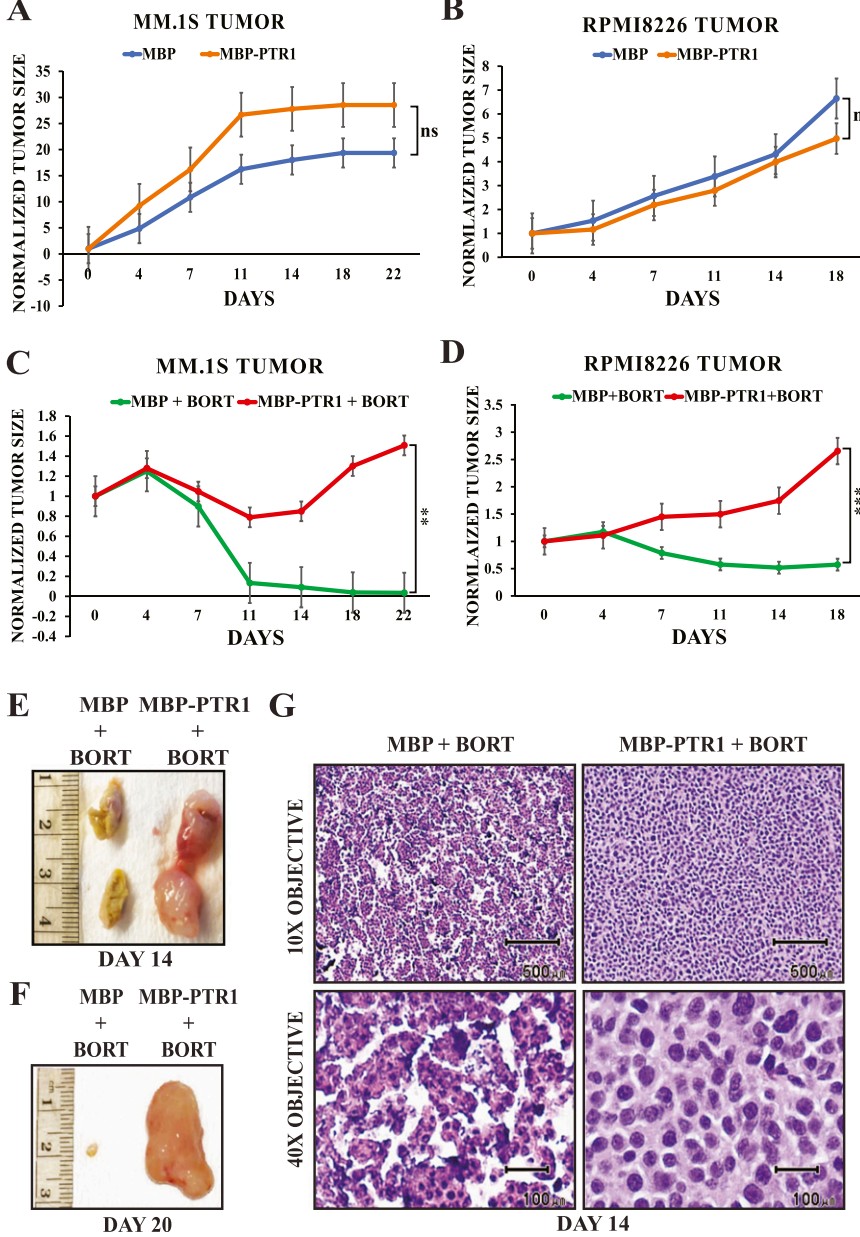

**Figure 2. MBP-PTR1 induces bortezomib resistance in MM cells in vivo.**
**(A, B)** MM.1S and (B) RPMI8226 cells were implanted in the flanks of NSG mice, and 100 μl of MBP-only control (425 ng/tumor) or MBP-PTR1 (500 ng/tumor) was intratumorally injected 1 d before each intraperitoneal injection of vehicle control (days 0, 4, 7, 11, and 14). Tumor responses are plotted as mean tumor volume ± SEM, normalized to baseline mean tumor volume at day 0. Area under the curve was calculated for each individual mouse, and the mean area under the curve of relevant treatment groups was compared using an unpaired *t* test analysis; ns, not significant. n = 3–4 mice for each treatment group. **(C, D)** MM.1S and (D) RPMI8226 cells were injected into NSG mice as above, and 100 μl of MBP-only control (425 ng/tumor) or MBP-PTR1 (500 ng/tumor) was intratumorally injected 1 d before each intraperitoneal injection of bortezomib (Bort at 1 mg/kg, days 0, 4, 7, 11, and 14). Tumor responses are plotted and analyzed as above. ***P* < 0.01 and ****P* < 0.001 (*t* test). **(E, F)** Representative bortezomib-treated MM tumors from mice injected with MBP-only (control) and MBP-PTR1 are resected to display their gross morphology at indicated days. Tumors in (E) were cut open, and both halves of tumors are displayed to show interiors of tumors. **(E, G)** H&E staining images of tumors shown in (E). H&E histology at low magnification (10×; scale bars: 500 μm) shows that MBP-only (control) tumor presents discohesive architecture in contrast to a confluent sheet of neoplastic tumor cells in MBP-PTR1 tumor. High magnification (40×; scale bars: 100 μm) shows the discohesive architecture of MBP-only (control) tumor is due to focal areas of cellular degeneration and necrosis that are absent in MBP-PTR1 tumor.

marked reduction in background signal (Figs 3C and S2A). In contrast, biotin-labeled MBP-only peptide failed to detect such a target band (Fig S2B).

Next, the biotinylated ~55-kD prey was enriched by streptavidin-agarose pulldown, resulting in a greater reduction in background signal (Fig 3D). Upon further analysis, it was found that $10^8$ RPMI8226 cells per biotinylation reaction yield optimal purity of ~55-kD target, and the pooling of samples from five separate parallel reactions was needed to visualize the target prey in a gel stained with GelCode Coomassie dye (Fig 3E). The ~55-kD target band and a similar area in the competition lane were excised from the stained gel, and liquid chromatography–tandem mass spectrometry (LC-MS/MS) analysis was performed to identify the putative 55-kD protein present. Although 28 MS/MS counts of CH60 peptides were

detected in the experimental group, the count of CH60 peptides reduced to 4 upon competition with excess unlabeled bait (Table S1A and B). The presence of CH60 after streptavidin-agarose pulldown was confirmed by anti-CH60 Western blotting, along with an expected reduction in the signal upon competition with excess unlabeled bait (Fig 3F). Using the same approach, we also detected CH60 with biotin-labeled MBP-PTR1 in the membrane fraction of MM.1R cells, but only a very low level of CH60 was detected in U266 cells (Fig S2C); as previously noted, U266 showed virtually undetectable NF-κB signaling upon MBP-PTR1 stimulation (Fig S1F). In addition, the cell lines that showed lower PTR1-mediated NF-κB responses (Fig S1D–F) also displayed commensurately lower levels of total CH60 expression (Fig S2D) compared with the robustly PTR1-responsive RPMI8226 cells (Fig 1F). Overall,

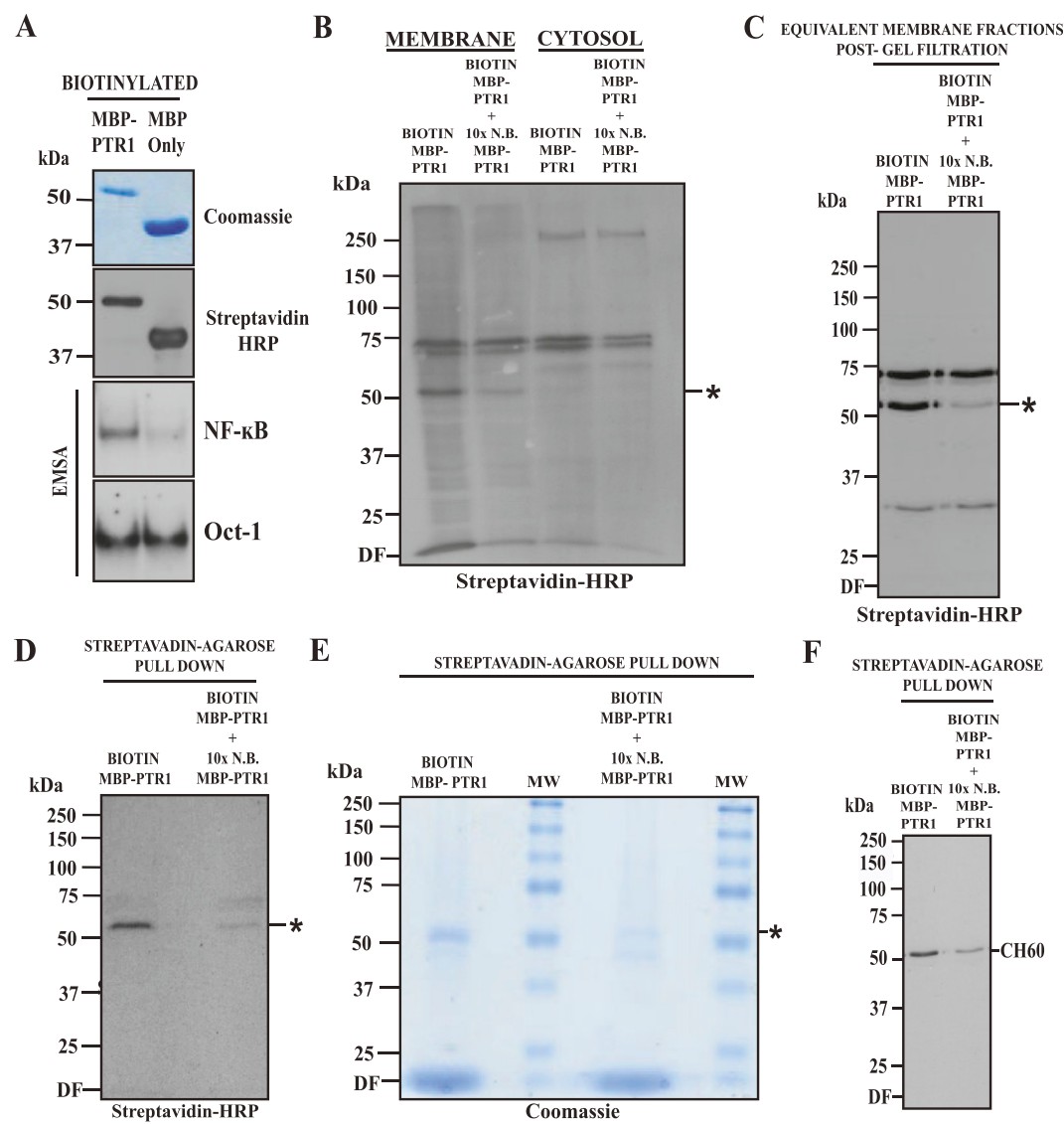

**Figure 3. PTR1 domain of HAPLN1 interacts with CH60 on the cell surface of MM cells.**
**(A)** Coomassie blue staining and streptavidin immunoblot of MBP-only and MBP-PTR1 peptide post-biotinylation with Sulfo-SBED tag. EMSA analysis of RPMI8226 cells treated with corresponding biotinylated peptides at 100 nM for 2 h. **(B)** Streptavidin immunoblot of equivalent RPMI8226 subcellular fractions after incubation of whole cells with 100 nM of biotinylated MBP-PTR1 ("Biotin MBP-PTR1"), in the absence or presence of 1 $\mu$M of non-biotinylated (N.B.) MBP-PTR1 (10× N.B. MBP-PTR1), for 15 min at 4°C, and subsequent photoactivation with UV light for 5 min. Biotinylated ~55-kD target band marked with "*." **(B, C)** Streptavidin immunoblot of equivalent RPMI8226 membrane fractions post-gel filtration of total membrane fraction from (B). **(C, D)** Streptavidin immunoblot of streptavidin-agarose pulldown samples from relevant gel filtration fractions, which show ~55-kD target band as seen in (C). Equivalent 10× N.B. MBP-PTR1 gel filtration fractions are also run in parallel. **(D, E)** Coomassie blue staining of pooled samples as in (D). **(D, F)** Immunoblot analysis of samples in (D) with anti-CH60 antibody. Results are representative of at least three independent biological replicates. HRP, horseradish peroxidase.

these results identified membrane-associated CH60 as the major direct interactor of HAPLN1-PTR1 in MM cells.

## CH60 is present on the plasma membrane and necessary for optimal NF-κB signaling elicited by HAPLN1-PTR1 in MM cells

CH60, also known as heat shock protein 60, normally localizes to the mitochondria, where it functions as a chaperone to facilitate stable conformation of polypeptides (Nisemblat et al, 2015). However, CH60 has also been reported to be ectopically localized at the plasma membrane of various mammalian cell types, including

human tumor cells, where it could evoke cellular behaviors ranging from phagocytosis to cell spreading (Soltys & Gupta, 1996, 1997; Barazi et al, 2002; Stefano et al, 2009). To independently assess the presence of CH60 on the surface of MM cells, immunofluorescence was performed on live, unpermeabilized RPMI8226 cells using a monoclonal anti-CH60 antibody (see details in the Materials and Methods section). Confocal and super-resolution imaging detected cell surface CH60 on MM cells as "patchy" puncta (Fig 4A) studded through the MM plasma membrane, which was uniformly stained using monoclonal anti-CD138 antibody, a bona fide positive control for cell surface localization in MM cells. An image of a mitotic cell

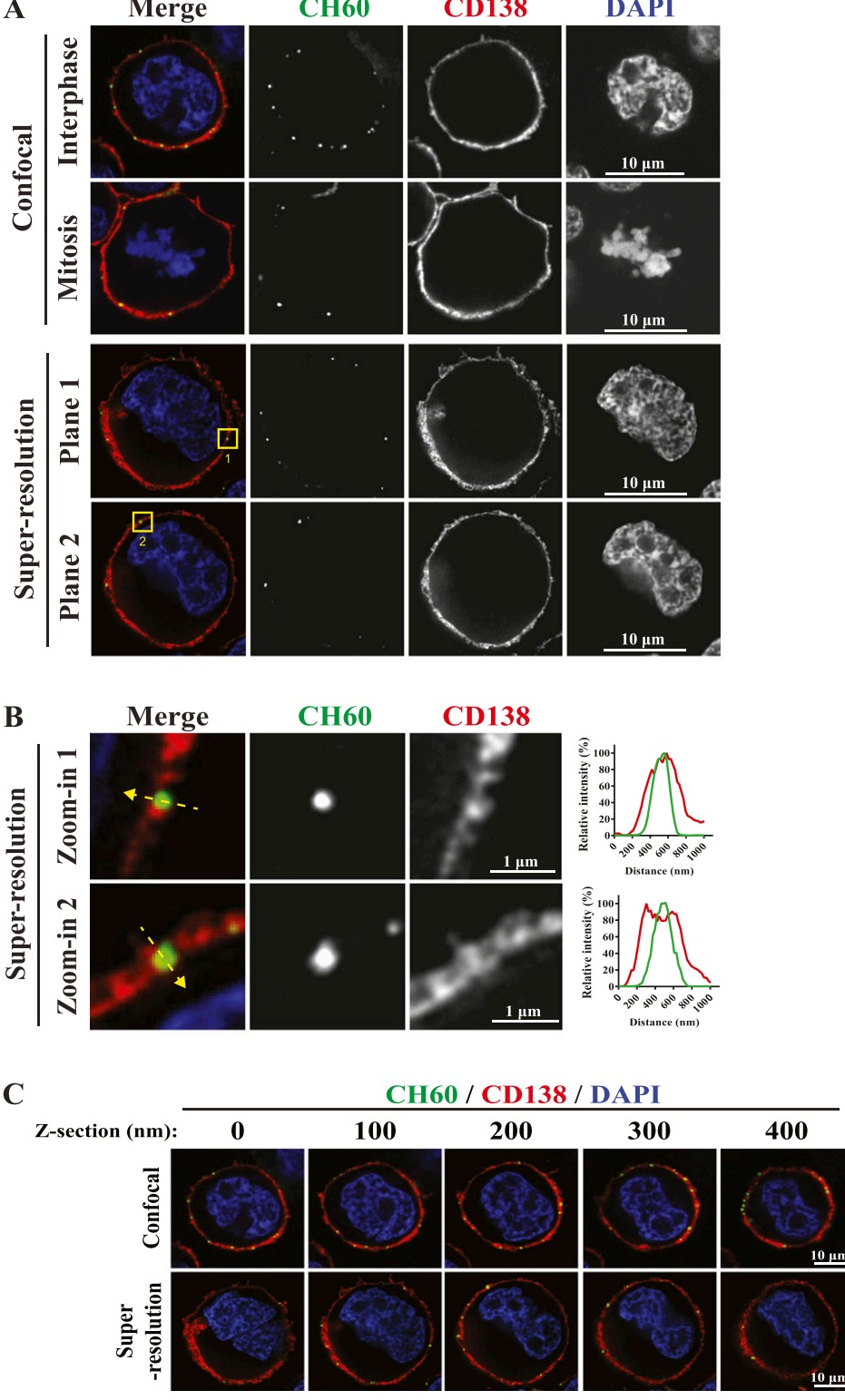

**Figure 4. CH60 is ectopically localized at the plasma membrane of MM cells.**
**(A)** Representative single Z-plane confocal and super-resolution images (scale bars: 10 μm) of intact, unpermeabilized RPMI8226 cells incubated with anti-CH60 antibody and anti-CD138 antibody. Cell surface CH60 (green) is observed as distinct foci studded on the plasma membrane. Uniform staining of CD138 (red), a cell surface marker for plasma cells, labels the plasma membrane. **(A, B)** Zoomed-in images (scale bars: 1 μm) of the super-resolution images from (A). Super-resolution images are used to generate intensity profile plots for CH60 (green) and CD138 (red) signals to observe degree of signal colocalization.
**(A, C)** Representative confocal and super-resolution images (scale bars: 10 μm) of five consecutive Z-planes of intact, unpermeabilized RPMI8226 cells as stained in (A).

was included to clearly depict the unstained cytoplasmic region in these unpermeabilized MM cells. Furthermore, intensity profile plots of CH60 and CD138 signals generated using super-resolution imaging (Fig 4B) showed clear visualization of the colocalization of these two proteins, suggesting that CH60 is aberrantly localized at the MM plasma membrane. Images of five consecutive sections along the Z-plane of the same cells as in Fig 4A reveal that multiple CH60 puncta are present in distinct locations at different planar

regions of the plasma membrane (Fig 4C). A 3D movie rendered using the different planes of an intact MM cell captured by super-resolution microscopy (Video 1) confirms that all CH60 puncta localize on or are proximal to the plasma membrane labeled by CD138 antibody. Preincubation with isotype control for anti-CH60 antibody failed to generate a signal compared with anti-CH60 staining (Fig S3A). Mean fluorescence intensity (MFI) analysis showed that anti-CH60 antibody staining generated a significantly stronger signal than with isotype control antibody (Fig S3B).

Next, to interrogate the role of CH60, we attempted to knock down CH60 gene expression in MM cells, as a complete knockout of CH60, an essential gene, would be lethal (Christensen et al, 2010). We titrated a lentiviral packaged shRNA construct targeting CH60 to achieve ~50% of WT expression in RPMI8226 cells (Fig 5A). Incubation with biotin-labeled MBP-PTR1 in the cell surface biotinylation assay resulted in a weaker biotinylated CH60 signal in the plasma membrane fraction of CH60 knockdown cells compared with control cells (Fig 5B). The reduced biotinylation signal observed in CH60 knockdown cells thus confirmed that the identity of ~55-kD target band detected by streptavidin HRP was indeed CH60. Furthermore, staining CH60 knockdown cells with anti-CH60 antibody generated a weaker signal compared with staining control cells (Fig S3C). MFI analysis showed that anti-CH60 antibody staining generated a significantly weaker signal in CH60 knockdown cells compared with in control cells (Fig S3D). The reduced MFI observed in CH60 knockdown cells thus confirmed that the identity of the "patchy" puncta detected by anti-CH60 antibody was indeed CH60.

CH60 knockdown RPMI8226 cells were then stimulated with different ligands to measure NF-κB response. Stimulation by TNF-α activated NF-κB at equivalent levels in both control and CH60 knockdown cells (Fig 5C and D). In contrast, CH60 knockdown cells showed a ~50% decrease in MBP-PTR1–induced NF-κB signaling compared with control cells (Fig 5C and D). A similar experiment was performed in MM.1S and MM.1R cells transfected with CH60-targeting siRNAs as lentiviral infection was not found to be optimal in these cells (Fig S4A and B). CH60 knockdown in both cell lines displayed a similar pattern of NF-κB signaling, with equivalent TNF-α–induced NF-κB activity and a significant reduction in MBP-PTR1–induced NF-κB activity compared with respective control cells (Fig S4C–F). These results demonstrated that CH60 knockdown specifically reduced HAPLN1-PTR1–induced NF-κB activity in MM cells.

### Cell surface CH60 mediates HAPLN1-PTR1–induced NF-κB signaling via TLR4 on the plasma membrane in MM cells

CH60 has been implicated to interact with innate immune receptors TLRs 2 and 4 in murine macrophages and dendritic cells (Vabulas et al, 2001). In addition, exogenously supplied CH60 in the culture media has been shown to signal through TLR4 to activate primary mouse B cells (Cohen-Sfady et al, 2005). TLR2 is generally not expressed in human MM cells, but TLR4 has been reported to be overexpressed in plasma cells from MM patients over those from normal healthy donors (Xu et al, 2010; Abdi et al, 2013). Thus, we next tested the hypothesis that cell surface CH60 acted in concert with TLR4 to mediate HAPLN1-PTR1–induced NF-κB signaling in MM cells.

First, we interrogated whether CH60 forms a complex with TLR4 on the surface of MM cells. Live, unpermeabilized RPMI8226 cells

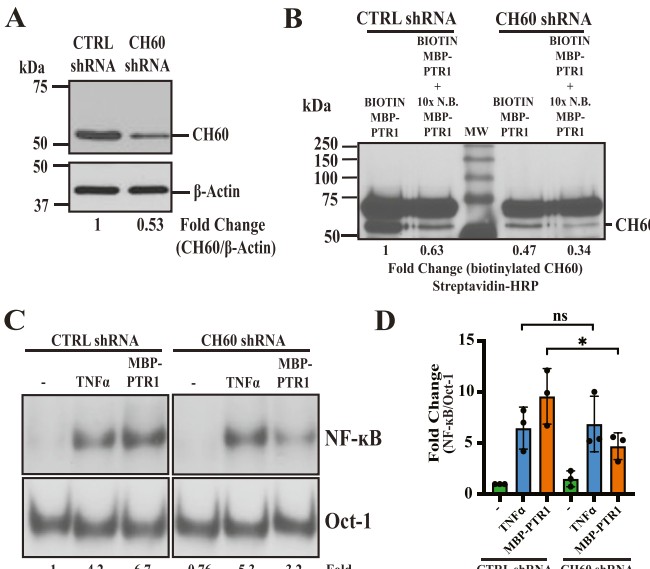

**Figure 5. CH60 is required for optimal HAPLN1-PTR1–induced NF-κB activity in MM cells.**
**(A)** Immunoblot analysis with anti-CH60 and anti–β-actin antibodies of RPMI8226 cells stably transduced with control (CTRL) or CH60 shRNAs. **(A, B)** RPMI8226 cells from (A) were incubated with biotinylated MBP-PTR1 as described earlier and subsequently subjected to subcellular fractionation and gel filtration. Streptavidin immunoblot of equivalent membrane fractions post-gel filtration of CTRL shRNA and CH60 shRNA RPMI8226 cells is shown. Relative fold change of CH60 signals is indicated below the blots. Immunoblots are representative of at least two independent biological replicates. **(A, C)** Representative EMSA analysis of RPMI8226 cells from (A) incubated with 100 nM of MBP-PTR1 for 2 h or incubated with 10 ng/ml TNF-α for 15 min. **(C, D)** Quantification of fold change of NF-κB activation relative to unstimulated control as in (C) from three independent experiments is plotted. Fold change of NF-κB activity was determined as previously described. ns, not significant; *P < 0.05 (t test).

were costained with anti-CH60 and anti-TLR4 antibodies for immunofluorescence analysis. Images of different Z-planes in super-resolution analyses showed that a subset of cell surface CH60 and TLR4 colocalized with each other in punctate-like complexes on the MM cell surface (Fig 6A). Furthermore, the intensity profile plots of the CH60 and TLR4 signals displayed clear visualization of the colocalization of these two proteins (Fig 6A). In addition, stimulation of RPMI8226 cells with MBP-PTR1 did not alter the frequency of colocalization between CH60 and TLR4, with ~65% of the CH60 puncta and ~60% of TLR4 puncta remaining colocalized before and after MBP-PTR1 treatment (Fig 6B and C). Interestingly, a proportion of both CH60 and TLR4 puncta was seen unassociated with each other on the plasma membrane, suggesting that they can exist independently at the myeloma cell surface. In addition, MBP-PTR1 stimulation did not alter the number of CH60 and TLR4 puncta present on the MM cell surface, with ~45 puncta of CH60 and ~50 puncta of TLR4 being present before and after MBP-PTR1 treatment (Fig 6B and D). Collectively, this suggests that a subset of CH60 and TLR4 exist as preformed complexes present on the MM cell surface whose colocalization and cell surface expression are independent of HAPLN1-PTR1 stimulation.

Next, we tested whether this interaction between CH60 and TLR4 at the MM cell surface was specific. When cell surface CH60 was biotin-labeled via biotinylated MBP-PTR1 as previously described,

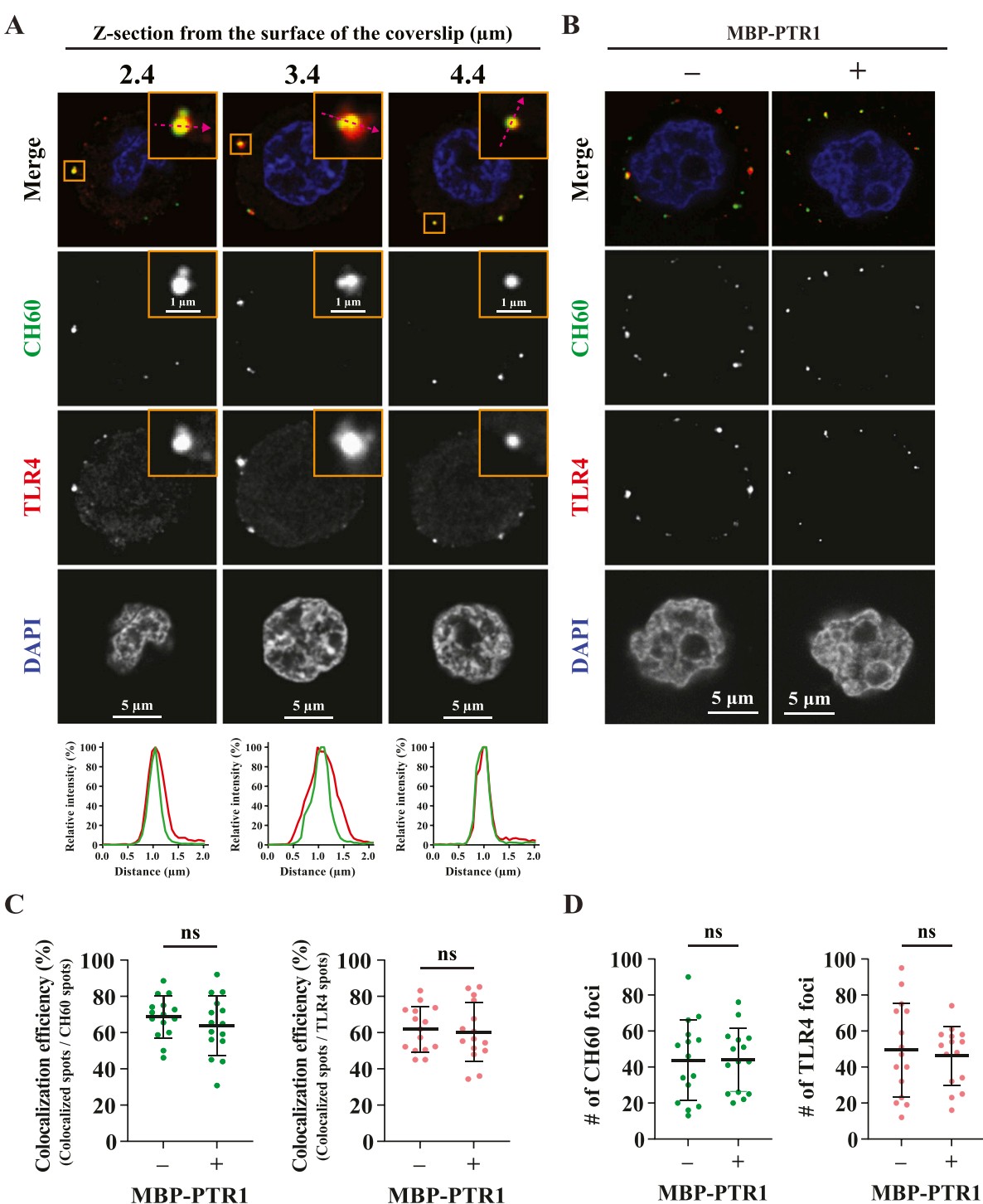

**Figure 6. CH60 colocalizes with TLR4 at the MM cell surface.**
**(A)** Representative Z-plane confocal images (scale bars: 5 μm) of intact, unpermeabilized RPMI8226 cells incubated with anti-CH60 antibody and anti-TLR4 antibody. Cell surface CH60 (green) and TLR4 (red) are observed as distinct foci studded on the plasma membrane. Zoomed-in images (scale bars: 1 μm) of the confocal images are used to generate intensity profile plots for CH60 (green) and TLR4 (red) signals to observe degree of signal colocalization. **(A, B)** Representative confocal images (scale bars: 5 μm) of RPMI8226 cells, which were pretreated with 100 nM of MBP-PTR1 for 30 min at 4°C as indicated, stained as in (A). **(B, C)** Efficiency of colocalization of CH60 and TLR4 puncta in (B) is plotted from three biological replicates as mean ± SD. Each dot represents colocalization efficiency (%) from a single cell. ns, not significant (*t* test). **(B, D)** Number of CH60 and TLR4 puncta in (B) is plotted from three biological replicates as mean ± SD. Each dot represents the number of foci from a single cell. ns, not significant (*t* test).

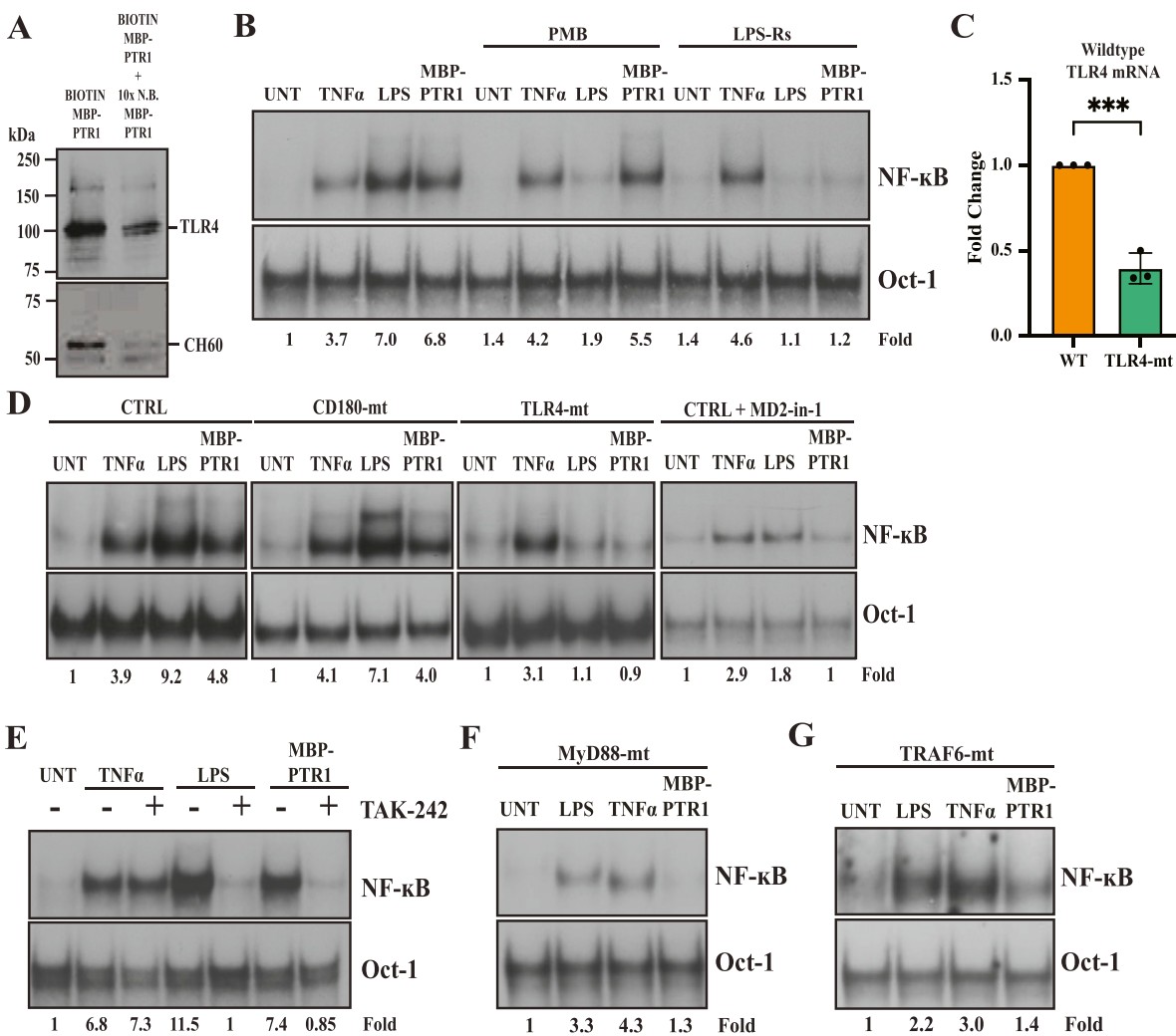

**Figure 7. TLR4 is necessary for HAPLN1-PTR1–induced NF-κB activity in MM cells.**
**(A)** Immunoblot analysis with anti-TLR4 and anti-CH60 antibodies of streptavidin-agarose precipitated equivalent membrane fractions post gel filtration as in Fig 3F is shown. Immunoblot is representative of at least two independent biological experiments. **(B)** EMSA analysis of RPMI8226 cells pretreated with 25 μg/ml of LPS-Rs or 10 μg/ml polymyxin B and then incubated with 10 ng/ml TNF-α for 15 min, 50 ng/ml LPS for 2 h, or 100 nM of MBP-PTR1 for 2 h **(C)** qRT–PCR was used to measure relative levels of WT TLR4 mRNA between control and TLR4-mutant (mt) RPMI8226 cells, normalized to GAPDH housekeeping gene. Results represent the mean ± SD of three biological replicates, each performed in triplicate. ***P < 0.001 (t test). **(D)** EMSA analysis of WT control (CTRL), CD180-mt, and TLR4-mt RPMI8226 cells pretreated with 20 μM of MD2-in-1 for 30 min as indicated and incubated with 10 ng/ml TNF-α for 15 min, 50 ng/ml LPS for 2 h, or 100 nM of MBP-PTR1 for 2 h **(E)** EMSA analysis of RPMI8226 cells pretreated with 10 μM of TAK-242 for 1 h as indicated and then incubated with 10 ng/ml TNF-α for 15 min, 50 ng/ml LPS for 2 h, or 100 nM of MBP-PTR1 for 2 h **(F)** EMSA analysis of MyD88-mt RPMI8226 cells incubated with 10 ng/ml TNF-α for 15 min, 50 ng/ml LPS for 2 h, or 100 nM of MBP-PTR1 for 2 h **(G)** EMSA analysis of TRAF6-mt RPMI8226 cells incubated with 10 ng/ml TNF-α for 15 min, 50 ng/ml LPS for 2 h, or 100 nM of MBP-PTR1 for 2 h. EMSA results are representative of at least three independent biological experiments. Fold change of NF-κB activity was determined as previously described.

streptavidin agarose pulled down both CH60 and TLR4, along with a concordant reduction in the signal upon competition with excess non-biotinylated MBP-PTR1 (Fig 7A). However, other surface proteins, such as CD14 and TLR9, were not pulled down (Fig S5A). These results suggested that CH60 specifically interacted with TLR4 in the plasma membrane fraction of MM cells. TLR4 recognizes its canonical ligand, bacterial LPS/endotoxin (Poltorak et al, 1998), and elicits a downstream signaling cascade that activates transcription factors such as NF-κB (Muroi et al, 2002). To test the role of TLR4 in HAPLN1-PTR1–induced NF-κB signaling, RPMI8226 cells were stimulated with the ligand in the absence or presence of a direct TLR4 antagonist, LPS-Rs (Anwar et al, 2015; Jurga et al, 2018). This inhibitor

abrogated MBP-PTR1–mediated NF-κB activity (Fig 7B). As expected, it also completely abrogated LPS-induced NF-κB activation while sparing TNF-α–induced activation. Moreover, LPS-Rs also abrogated HAPLN1-PTR1–induced NF-κB activity in MM.1S and MM.1R cells (Fig S5B and C). These data collectively suggest that HAPLN1-PTR1 activates NF-κB through a cell surface recognition protein, CH60, which then signals via associated TLR4.

To further test the role of TLR4, CRISPR/Cas9 was employed to target a functionally important N-terminal domain of TLR4 using gRNA delivered into RPMI8226 cells. This resulted in a genetically edited TLR4-mutant cell pool (TLR4-mt) comprising a mixed population of cells possessing different non-functional TLR4 mutants

that shared a common gRNA-directed cut site based on the bio-informatics tool, Inference of CRISPR Edits (ICE; Synthego) (Fig S5D). Comparison of Sanger sequencing data between control and TLR4-mt cell pools indicated a high rate of induced insertion–deletion (indel) mutations, with 96% of the edited TLR4-mt cell pool possessing mutant TLR4. Furthermore, using the shared gRNA target cut site as the forward primer in a qRT–PCR primer pair, Quantitative RT–PCR (qRT–PCR) revealed that the TLR4-mutant cell pool had significantly less mRNA transcript levels corresponding to WT TLR4, as compared to unedited control RPMI8226 cells (Fig 7C). We also generated a similar non-functional CD180-mutant cell pool (CD180-mt) as a negative control (Fig S5E), as CD180 is a TLR family member closely related to TLR4, activated by LPS/endotoxin, and overexpressed in patient MM cells (Kikuchi et al, 2018). CD180-mt continued to activate NF-κB in response to MBP-PTR1, and to TNF-α and LPS, similar to unedited control cells (Fig 7D, first and second panels). In contrast, the TLR4-mutant cell pool was unable to respond to MBP-PTR1 and to LPS, but retained their ability to signal NF-κB in response to TNF-α (Fig 7D, third panel). In agreement with these data, we also observed that a similarly generated TLR4-mutant MM.1R cell pool (Fig S5F) was also incapable of mounting NF-κB response upon MBP-PTR1 treatment (Fig S5G).

To further confirm that the MBP-PTR1–induced NF-κB response is dependent on TLR4 signaling, we interrogated the role of known mediators of the TLR4 signal cascade. When cell surface CH60 was biotin-labeled via biotinylated MBP-PTR1 as previously described, streptavidin agarose pulled down MD-2 (Fig S5A), a cell surface protein well described to be physically associated with TLR4 (Shimazu et al, 1999). Consistent with this finding, RPMI8226 cells pretreated with MD2-in-1, a chemical inhibitor of MD-2 (Zhang et al, 2016), abrogated MBP-PTR1–induced NF-κB activity (Fig 7D, fourth panel). Similarly, pretreatment of RPMI8226 cells with TAK-242, a small-molecule inhibitor of TLR4 signaling that blocks its interaction with downstream adaptor molecules such as MyD88 (Matsunaga et al, 2011), selectively abrogated NF-κB responses mediated by MBP-PTR1 and LPS, but left TNF-α–induced NF-κB signaling intact (Fig 7E). We also generated CRISPR-edited MyD88 and TRAF6-mutant RPMI8226 pools (Fig S5H and I) to further test the role of TLR4 in HAPLN1-PTR1 signaling, because these two proteins are known to act downstream of TLR and upstream of NF-κB (Lu et al, 2008). Consistent with the role of TLR4 in signaling to NF-κB in response to HAPLN1-PTR1, these MyD88- and TRAF6-mutant (MyD88-mt and TRAF6-mt) cells were also defective in MBP-PTR1–induced NF-κB signaling (Fig 7F and G). Collectively, chemical inhibitor and genetic mutant analyses indicate that cell surface CH60 directly recognizes HAPLN1-PTR1 and signals via a TLR4/MyD88/TRAF6 cascade to cause NF-κB activation in MM cells.

### HAPLN1-PTR1 requires TLR4 to induce survival genes and promote drug resistance in MM cells

Because NF-κB generally induces drug resistance by activating transcription of pro-survival genes, we next assessed TLR4-dependent induction of known NF-κB–regulated cell survival genes in response to MBP-PTR1 stimulation in MM cells. When RPMI8226 cells were treated with MBP-PTR1 for 6 h, induction of several NF-κB–regulated anti-apoptotic genes was observed, such

as *BCL2A1* (989.02-fold), *BIRC3* (40.46-fold), and *BCL2* (5.12-fold), along with pro-survival cytokines, such as *CXCL8* (474.63-fold) and *IL10* (489.33-fold) (Fig 8A and B). In contrast, TLR4-mutant RPMI8226 cells displayed marked reductions in these pro-survival genes after stimulation with MBP-PTR1, consistent with the loss of MBP-PTR1–induced NF-κB activation (Fig 8A and B). Furthermore, in the absence of MBP-PTR1 stimulation, TLR4-mutant RPMI8226 cells showed negligible differences in the levels of these NF-κB–regulated genes compared with their WT counterparts (Fig S5J). This suggests that TLR4 mutation does not basally cause any potent transcriptional changes in RPMI8226 cells.

To test whether TLR4 also mediates drug resistance induced by HAPLN1-PTR1 in MM cells, cell viability was tested in the presence or absence of the clinical drug bortezomib. MBP-PTR1 did not change the viability of RPMI8226, MM.1S, and MM.1R cells in the absence of bortezomib, but it induced bortezomib resistance to these MM cells to variable degrees (Fig 8C–E). Importantly, TLR4 deficiency caused complete loss of bortezomib resistance induced by MBP-PTR1 in RPMI8226 and MM.1R cells (Fig 8F and G). Collectively, these results demonstrate that TLR4 is necessary for HAPLN1-PTR1 to induce NF-κB–regulated cell survival genes and promote drug resistance in MM cells.

## Discussion

In the present study, we generated recombinant HAPLN1-PTR1 that exhibited improved quality and functional potency, and used it to identify critical components of a putative receptor complex through which a HAPLN1 matrikine elicits pro-tumorigenic signaling in MM cells. An unbiased cell surface biotinylation assay and mass spectrometric analysis identified CH60 as a candidate direct binding interactor of HAPLN1-PTR1 on MM cell surface membrane. We visualized through confocal and super-resolution imaging that CH60 is present at the MM cell surface, confirming its extra-mitochondrial localization. Genetic knockdown of CH60 attenuated its interaction with HAPLN1-PTR1 and inhibited HAPLN1-PTR1–induced NF-κB activity. CH60 was shown to physically associate with TLR4 at the MM cell surface, and TLR4 was determined to be necessary to evoke downstream NF-κB signaling, transcription of pro-survival genes, and bortezomib resistance in MM cells. Together, these results show that HAPLN1-PTR1 signals through a novel CH60-TLR4 cell surface complex to activate pathobiological NF-κB signaling and mediate drug resistance in MM cells. Our work provides new mechanistic insight into how MM cells use a previously undescribed receptor complex to co-opt a matrikine derived from the structural ECM protein HAPLN1 as a cancer-promoting signaling element.

CH60 is traditionally considered a mitochondrial chaperone protein and a common essential gene (Perezgasga et al, 1999; Christensen et al, 2010). However, it is reported to aberrantly localize to the plasma membrane when cells experience a variety of cellular stress including carcinogenesis (Laad et al, 1999; Shin et al, 2003; Cicconi et al, 2004; Campanella et al, 2012), infection (Belles et al, 1999), and inflammation (Xu et al, 1994). Of particular relevance to our study, primary myeloma cells from MM patient BM have been described to aberrantly express cell surface CH60, in contrast to

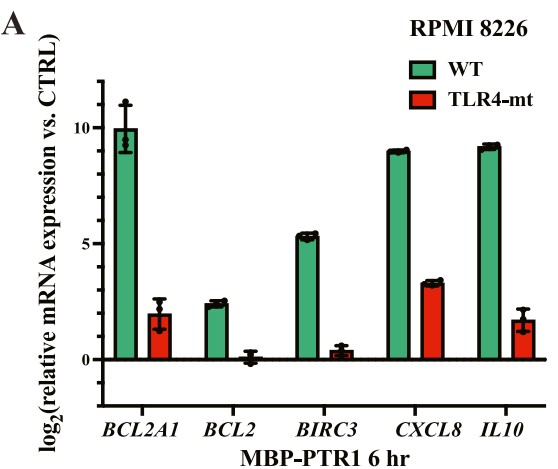

**B**

| Gene Name | RPMI 8226 WT (Fold Change) | RPMI 8226 TLR4-mt (Fold Change) |
|---|---|---|
| *BCL2A1* | 989.02 | 3.89 |
| *BCL2* | 5.12 | 1.06 |
| *BIRC3* | 40.46 | 1.32 |
| *CXCL8* | 474.63 | 9.38 |
| *IL10* | 489.33 | 1.74 |

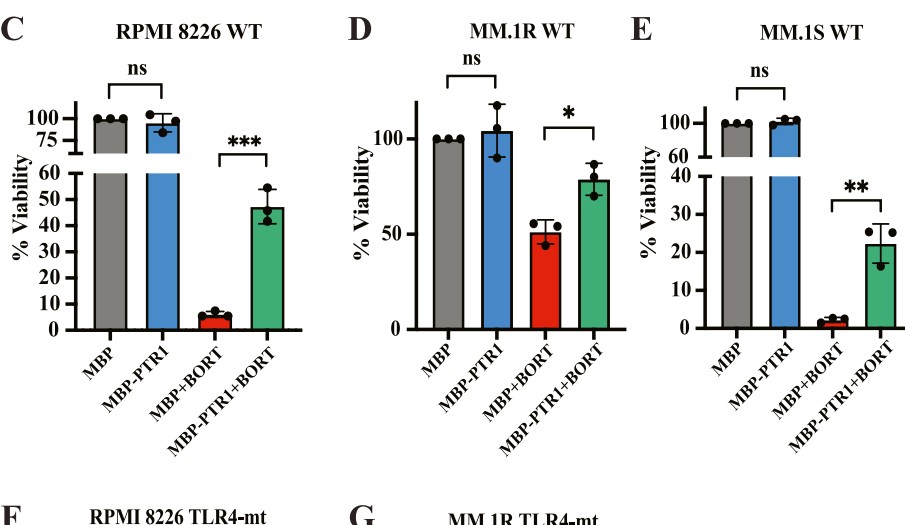

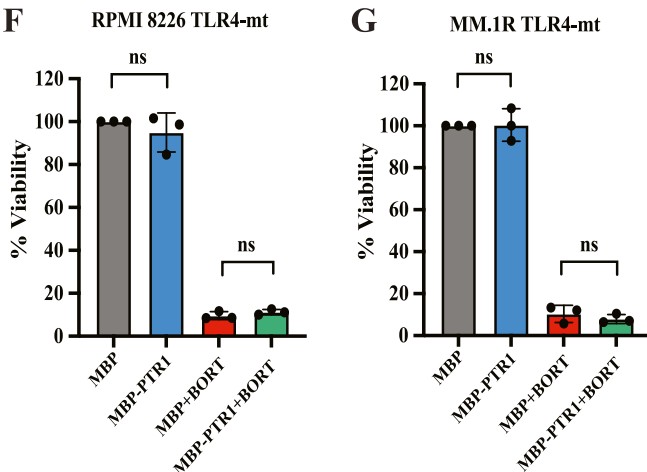

**Figure 8. TLR4 is required for HAPLN1-PTR1 to promote drug resistance in MM cells.**
**(A)** qRT–PCR analysis of NF-κB regulated pro-survival genes between WT and TLR4-mt RPMI8226 cells treated with either 100 nM of MBP-only (control) or MBP-PTR1 for 6 h. RNA levels of indicated genes in MBP-PTR1–treated cells were normalized to GAPDH housekeeping gene, and log$_2$ transformation of fold change relative to control (MBP-only) for each gene is plotted. The x-axis represents the basal RNA level corresponding to control treatment. Results represent the mean ± SD of three biological replicates, each performed in triplicate. **(A, B)** Raw fold change in RNA induction of indicated genes between WT and TLR4-mt RPMI8226 cells as treated and analyzed in (A). **(C)** RPMI8226 cells were cultured at 5 × 10$^5$ cells in a 24-well plate with 100 nM MBP-only (control) or 100 nM MBP-PTR1 alone or in combination with 5 nM bortezomib (Bort) for 48 h **(D, E)** MM.1S and (E) MM.1R cells were cultured at 5 × 10$^5$ cells in a 24-well plate with 200 nM MBP-only (control) or 200 nM MBP-PTR1 alone or in combination with 1 nM bortezomib. After 48 h, cell viability was measured by trypan blue exclusion assay. Each dot represents the mean of a biological replicate performed in triplicate. Bar graphs represent the mean of three biological replicates ± SD. ns, not significant; *$P$ < 0.05; **$P$ < 0.01; and ***$P$ < 0.001 ($t$ test). **(C, D, F, G)** Same cell viability

healthy BM donor cells (Rahlff et al, 2012). Interestingly, treatment with the drug bortezomib, a clinical mainstay of MM first-line therapy, was also shown to up-regulate surface exposition of CH60 in murine ovarian tumors (Chang et al, 2012). Cell surface CH60 is also described to act as an agonist of various receptors, such as $\alpha3\beta1$ integrin on breast cancer cells (Barazi et al, 2002), TREM2 on microglia (Stefano et al, 2009), and CD18 on macrophages (Long et al, 2003). Although exogenously added CH60 has been described to cause cell signaling in a TLR4-dependent manner (Hoshino et al, 1999; Vabulas et al, 2001; Cohen-Sfady et al, 2005), to our knowledge, CH60 has not been previously described as a direct receptor for any extracellular ligand to modulate downstream signaling in any cell system. Furthermore, CH60 and TLR4 had not been shown to colocalize and exist as preformed complexes on the plasma membrane. Hence, our identification of CH60 as a novel coreceptor that exists on the plasma membrane to mediate matrikine-induced NF-$\kappa$B signaling and drug resistance in MM cells is unprecedented and warrants further investigation to assess its potential as a biomarker for TME-mediated drug resistance and MM disease progression.

TLR4 is also reported to be intimately involved in cancer progression (Kashani et al, 2021), and recent work shows that it may play an important role in MM pathogenesis. TLR4 present on both MM cells and MM mesenchymal stromal cells has been shown to promote MM cell proliferation, immune evasion, and drug resistance (Xu et al, 2010; Bao et al, 2011; Bagratuni et al, 2019; Giallongo et al, 2019, 2020; Lemaitre et al, 2022; Scandura et al, 2022). TLR4 is described to enact this pathobiological role by up-regulating mediators of inflammation, decreasing unfolded protein response, and bypassing drug-induced oxidative stress. One of the primary challenges in studying the role of TLR4 in MM disease is being able to stimulate TLR4 signaling in a manner that is physiologically relevant to MM pathology. The above studies used bacterial LPS/endotoxin as an artificial surrogate to modulate TLR4 activity in MM cells. Our data suggest that HAPLN1 matrikine acts through a CH60-TLR4 complex lacking CD14, a critical coreceptor in LPS-induced signaling (Wright et al, 1990; Muroi et al, 2002), thus distinguishing it from the classical TLR4 complex engaged by LPS. These results provide the initial evidence that a non-classical TLR4 complex is activated by a matrikine endogenously enriched in certain MM patients and further credence to the rationale that therapeutically targeting TLR4 may help inhibit MM progression.

HAPLN1 has been previously reported to promote tumorigenesis with no clear mechanistic understanding. For example, high HAPLN1 expression in lung tumors of mesothelioma patients (Ivanova et al, 2009), liver tumors of hepatocellular carcinoma patients (Mebarki et al, 2016), and triple-negative breast cancer patients with a TME that promoted immune evasion (Kok et al, 2022) positively correlated with shorter overall survival (OS). Similarly, gastric cancer patients with high HAPLN1 levels in their cancer-associated fibroblasts showed shorter OS (Zhang et al, 2022a). The latter cell extrinsic up-regulation of HAPLN1 is in alignment with our laboratory's finding that mesenchymal stromal cells of the MM TME

secrete HAPLN1 and process it to a matrikine to promote MM progression (Mark et al., 2022). Computational analyses from triple-negative breast tumors and colorectal tumors also identified high HAPLN1 expression with the regulation of inflammatory and immune cytokines as part of its pathobiology (Yau et al, 2010; Zhang et al, 2022b). However, the exact mechanism through which HAPLN1 engenders a pro-inflammatory TME, or why its high expression is associated with poor OS of patients with different cancer types remains unclear. The identification of a CH60-TLR4 complex as a HAPLN1 matrikine "sensor" that signals to mediators of inflammation, such as NF-$\kappa$B, provides mechanistic insights into its pathobiological abilities and potential therapeutic targets to allay its pro-cancer contribution while preserving its native ECM function. The preservation of normal ECM function of HAPLN1 may be of importance as down-regulation of HAPLN1 could destabilize the ECM making it more permeable to cancer invasion of melanoma (Ecker et al, 2019; Kaur et al, 2019) and colorectal cancer (Wang et al, 2021). It is therefore of great interest to examine the potential role of overexpressed HAPLN1 as a matrikine, rather than an ECM structural element, in other cancer types and whether these malignancies also employ a CH60/TLR4 complex to detect a HAPLN1 matrikine to promote pro-tumorigenic properties.

# Materials and Methods

### Cell lines, antibodies, and chemicals

RPMI8226, MM.1S, MM.1R, and U266 MM cell lines (American Type Cell Culture) were cultured in 5% $CO_2$ incubators at 37°C, using Roswell Park Memorial Institute 1640 media (R7388; Sigma-Aldrich) supplemented with 10% FBS, 2% GlutaMAX (35050061; Gibco), and 1% penicillin/streptomycin. HEK293T cells (American Type Cell Culture) were cultured in 10% $CO_2$ incubators at 37°C using DMEM supplemented with 10% FBS and 1% penicillin/streptomycin. The authenticity of the above human cell lines was confirmed by short tandem repeat analysis performed independently by the Translational Research Initiatives in Pathology laboratory at UW Madison. RPMI8226- and MM.1R-mutant (mt) cell pools for CD180, TLR4, MyD88, and TRAF6, along with the matched WT control cell pool, were commercially generated by Synthego. Oct-1 probe (2506), antibodies reactive to $\beta$-actin (I-19), TLR4 (25), HAPLN1 (K-14, C-14), cRel (C), p65 (C-20), RelB (C-19), and normal mouse IgG2a (3878) were obtained from Santa Cruz Biotechnology. Monoclonal antibody raised against CH60 (4B9/89) was purchased from Invitrogen. Polyclonal antibody raised against TLR4 (SPC-200) used for immunofluorescence was purchased from StressMarq Biosciences. Monoclonal antibody raised against CD138 (EPR6454) was purchased from Abcam. Antibody reactive to $\alpha$-tubulin (DM1A) was purchased from EMD Millipore. Antibodies reactive to phospho-IKK$\alpha$/$\beta$ (2697), I$\kappa$B$\alpha$ (L35A5), and anti-mouse secondary antibody conjugated to Alexa Fluor 594 (8890) were purchased from Cell Signaling Technology. Reagents purchased include recombinant

assay as described in (C, D) was performed in TLR4-mt RPMI8226 and MM.1R cells, respectively. Each dot represents the mean of a technical replicate performed in triplicate. Bar graphs represent the mean of three biological replicates ± SD. ns, not significant (t test).

human TNF-α (654205; EMD Millipore), LPS (L2880; Sigma-Aldrich), LPS-Rs (tlrl-rslps; InvivoGen), bortezomib (S1013; Selleckchem), polymyxin B sulfate salt (P4932; Sigma-Aldrich), MD2-in-1 (S6573; Selleckchem), TAK-242 (S7455; Selleckchem), IKK16 (S2882; Selleckchem), and cycloheximide (C7698; Sigma-Aldrich).

### Generation of MBP-tagged fusion peptides

A gene segment encoding the human HAPLN1-PTR1 domain was purchased as a gBlocks gene fragment (Integrated DNA Technologies) with BamH1 and Xho1 cut sites engineered at the 5' end and 3' end, respectively. The dsDNA sequence was introduced into the pET28-MBP-TEV plasmid (69929; Addgene) such that the HAPLN1-PTR1–encoding sequence was in frame with MBP-encoding sequence and the resulting plasmid was transformed into Rosetta(DE3) E. coli cells (70954; Novagen). Cells were cultured in modified lysogeny broth supplemented with glucose (10 g/l Bacto Tryptone, 5 g/l yeast extract, 5 g/l NaCl, and 2 g/l glucose) at 37°C and gently agitated to an optical density (O.D.) of 0.6 at 600 nm. Protein production was then induced with 1 mM isopropyl β-D-1-thiogalactopyranoside (l6758; Sigma-Aldrich) at 20°C for 5 h. The cells were centrifuged, and the resulting cell pellet was resuspended and lysed via ultrasonication in ice-cold column buffer (200 mM NaCl, 20 mM Tris–HCl, 1 mM EDTA and 10% glycerol, pH 7.4). The crude lysate was centrifuged at 20,000$g$ to remove debris, and clarified lysate was diluted fivefold with column buffer and passed through equilibrated amylose resin (E8021S; New England Biolabs) using a gravity flow setup at 4°C. The resin-bound MBP-fusion peptide was washed overnight in column buffer, then eluted using 50 mM maltose in column buffer. Eluted peptide was stripped of bacterial endotoxin contamination using Pierce high-capacity endotoxin removal spin columns (88274; Thermo Fisher Scientific) as per vendor's instructions. The resulting eluate was subsequently injected into a benchtop ÄKTA FPLC attached to a Superose 6 10/300 Gl gel filtration column to separate the monomeric target moiety from smaller degraded peptides and incomplete translation products, along with some contaminants. Fractions with the monomeric peptide were then pooled, exchanged into loading buffer (20 mM Hepes and 2 mM EDTA, pH 7.4), and passed through a 1.0-ml MonoQ 5/50 Gl (GE Life Sciences) anion-exchange column to bind cationic molecules. Bound material was then eluted using a linear salt gradient from 0.0 to 1.0 M NaCl in loading buffer to enrich for the recombinant monomeric MBP-PTR1 peptide. Small samples of individual column fractions were electrophoresed in 10–15% SDS–PAGE, which was then incubated for 1 h in GelCode Coomassie blue safe protein stain (24594; Thermo Fisher Scientific), followed by destaining in deionized $H_2O$ overnight to visualize target bands. The final pooled MonoQ samples containing MBP-PTR1 were kept frozen in small aliquots at –80°C in storage buffer (200 mM NaCl, 20 mM Hepes, 1 mM EDTA, and 10% glycerol, pH 7.4).

### EMSAs

NF-κB activation in MM cell lines was measured by EMSA using [32]P end-labeled double-stranded DNA probes for NF-κB (5'-CTCAA-CAGAGGGGACTTTCCGAGAGGCCAT-3') and Oct-1 loading control (5'-TGTCGAATGCAAATCACTAGAA-3') with total cell extracts as previously described (Huynh et al, 2018). The EMSA gels were dried and exposed to X-ray films. The dried gels were also exposed to phosphor-image screens, and NF-κB and Oct-1 DNA-binding activities were imaged and quantified using a Typhoon scanner and the Image-Quant software (GE Healthcare). NF-κB activation in each sample was normalized to Oct-1 binding value from the same sample, and the fold change in NF-κB activation in experimental groups (NF-κB/Oct-1) was normalized to vehicle-treated control values (NF-κB/Oct-1) from the same sample. Where indicated, EMSA was done in the presence of a RelB antibody (RelB Aby) to super-shift basal RelB complexes to enable quantification of canonical NF-κB activation.

### SDS–PAGE and immunoblot analysis

Myeloma cell lines were pelleted and lysed in TOTEX buffer, as previously described (O'Connor et al, 2004). Equal amounts (20 µg) of soluble protein were run on denaturing 10–12.5% SDS–PAGE and transferred onto a polyvinylidene fluoride or nitrocellulose membrane (GE Healthcare). The membrane was then incubated with the appropriate antibodies as described previously (Miyamoto et al, 1998). Immunoblots were analyzed by enhanced chemiluminescence as described by the manufacturer (GE Healthcare). Quantification of immunoblots was performed using ImageJ (NIH) to calculate the signal intensity of protein(s) of interest normalized to total protein loading (actin) for each lane from the same sample and then to vehicle control values for each experiment.

### Biotin transfer assay to specifically tag direct binding partner of the HAPLN1-PTR1 domain

Recombinant MBP-PTR1 peptide generated as described above was labeled using the Sulfo-SBED biotin label transfer kit (33073; Thermo Fisher Scientific) and desalted as per vendor's instructions. 100 nM of this biotin-tagged "bait" peptide was incubated with $10^8$ MM cells in ice-cold complete media and mixed with end-to-end tumbling for 30 min at 4°C. In a separate reaction, 100 nM of biotin-tagged bait was also incubated with 10× excess unlabeled bait under the same conditions. The reaction was transferred to a shallow 60-mm petri dish on ice, and exposed to UV light at 365 nm from the OmniCure S1000 lamp for 5 min at a distance of 8 cm. The cells were washed in ice-cold PBS twice and then subjected to subcellular fractionation (Yu et al, 2013). Briefly, cells were agitated and pelleted in a subcellular fractionation buffer (250 mM sucrose, 20 mM Hepes (pH 7.4), 10 mM KCl, 1.5 mM $MgCl_2$, 1 mM EDTA, and 1 mM EGTA), followed by centrifugation of the supernatant at 10,000$g$ to remove the mitochondrial component. Clarified supernatant was then subject to ultracentrifugation at 100,000$g$ to pellet the plasma membrane component. The plasma membrane pellet post-ultracentrifugation was resolubilized in 500 µl of ice-cold TOTEX buffer. The membrane fraction was then centrifuged at 13,000$g$ at 4°C, and the supernatant was injected into a ÄKTA FPLC attached to a Superose 6, 10/300 Gl gel filtration column. Fractions were collected, and total protein content for each was determined. Equivalent protein amounts from chosen membrane fractions were electrophoresed in 10–15% SDS–PAGE under reducing conditions,

immunoblotted, and developed using streptavidin HRP as described above to visualize biotinylated binding partner.

## Streptavidin-agarose enrichment of biotinylated "prey" for LC-MS/MS analysis

Pierce streptavidin agarose (20353; Thermo Fisher Scientific) and Pierce control agarose (26150; Thermo Fisher Scientific) resins were equilibrated in ice-cold binding buffer (150 mM NaCl, 20 mM Tris–HCl, and 1 mM EDTA). The control resin was first introduced into pooled gel filtration fractions containing biotinylated prey. The slurry was mixed end-to-end at 4°C for 2 h to preclear the samples. The beads were gently pelleted, and the supernatant was transferred to a new tube containing streptavidin resin. The slurry was mixed end-to-end at 4°C overnight to bind and isolate the biotinylated prey protein. The resin was washed twice in ice-cold binding buffer with 0.2% Tween-20, boiled in Laemmli sample buffer, electrophoresed in 10–15% SDS–PAGE, and immunoblotted as described above. Based on preliminary analyses, it was deemed necessary to pool gel filtration fractions containing biotinylated prey from five separate independent experiments for LC-MS/MS analysis. The pooled resins were then subjected to SDS–PAGE and stained with GelCode Coomassie dye.

## In-gel tryptic digestion of Coomassie-stained target bands

Experiments were performed with the previously described protocol (Shevchenko et al, 2006). Briefly, after destaining the Coomassie gel, the target bands were excised into cubes (1 × 1 mm), and the resulting gel pieces were transferred into a microcentrifuge followed by dehydration with 500 $\mu$l neat acetonitrile. An additional reduction/alkylation step was performed. Freshly prepared 15 mM dithiothreitol was added into gel pieces and incubated for 30 min in a water bath at 56°C. After cooling to room temperature, the samples were dehydrated again as previously performed. Then, an equal volume of 55 mM iodoacetamide was added in the dark for 30 min. For the protein digestion, 15 ng/$\mu$l trypsin was added into the gel pieces and incubated overnight. The tryptic peptides were extracted by adding 5% formic acid (FA) solution. Samples were dried and stored at –80°C until use.

## LC-MS/MS analysis

The LC-MS/MS detection system consisted of a nanoflow HPLC instrument (Dionex UltiMate 3000 UPLC System; Thermo Fisher Scientific) coupled to a Q Exactive HF mass spectrometer (Thermo Fisher Scientific) with a nanoelectrospray ion source (Thermo Fisher Scientific). In brief, 0.5 $\mu$g of peptide mixture dissolved in buffer A (0.1% FA) was loaded onto a 75 $\mu$m × 15 cm fabricated column, which was filled with 1.7-$\mu$m Ethylene Bridged Hybrid packing materials (130 Å; Waters), over a 126-min linear gradient of 3–45% mobile phase B (buffer A, 0.1% FA in water; buffer B, 0.1% FA in ACN) at a flow rate of 300 nl/min. The MS analysis was performed in a data-dependent manner using an Orbitrap mass analyzer. For a full mass spectrometry survey scan, the target value was 1 × 10$^5$ and the scan ranged from $m/z$ 300 to 1,500 at a resolution of 60,000 and a maximum injection time of 100 ms. For the MS2 scan, up to 15 of the most intense precursor ions from a survey scan were selected for MS/MS and detected by the Orbitrap at a mass resolution of 15,000 at $m/z$ 400. Only spectra with a charge state of 2–6 were selected for fragmentation by higher energy collision dissociation with a normalized collision energy of 30%. The automatic gain control for MS/MS was set at 8e3, with maximum ion injection times of 100 ms. Dynamic exclusion time was 45 s, and the window for isolating the precursors was $m/z$ 1.4.

## Label-free–based MS quantification for proteins

Protein identification and quantification was carried out by MaxQuant (version 1.5.3.8)-based database searching, using the integrated Andromeda search engine with FDR < 1% at the peptide and protein level. The tandem mass spectra were searched against the human UniProt database (version 20200219; 20,193 sequences). A reverse database for the decoy search was generated automatically in MaxQuant. Enzyme specificity was set as "Trypsin," and a minimum number of seven amino acids were required for peptide identification. For label-free protein quantification (LFQ), the MaxQuant LFQ algorithm was used to quantitate the MS signals, and the proteins' intensities were represented in LFQ intensity (Cox et al, 2014). Cysteine carbamidomethylation was set as the fixed modification. The oxidation of M- and acetylation of the N-terminal proteins were set as variable modifications. The first search mass tolerance was 20 ppm, and the main search peptide tolerance was 4.5 ppm. The false discovery rates of the peptide-spectrum matches and proteins were set to less than 1%.

## Immunofluorescence for super-resolution and confocal imaging to measure colocalization

MM cells were incubated in media containing 10% FBS and Human TruStain FcX (422301; BioLegend), a Fc receptor blocking solution, for 30 min at 4°C. Cells were then washed in PBS and incubated with relevant antibodies diluted in PBS containing 3% BSA and 0.04% NaN$_3$ for 1 h at room temperature, followed by PBS washes. The cells were then fixed with 4% PFA and then cytospun onto the coverslip using a centrifuge (Shandon Cytospin; Thermo Fisher Scientific). Spun cells were fixed with 4% PFA followed by PBS wash. Subsequently, cells were stained with secondary antibodies conjugated with Alexa Fluor 488 or Rhodamine Red-X (Jackson ImmunoResearch) in a humidified chamber. Cells were washed with PBS and were stained with DAPI before mounting by homemade glycerol-based mounting media.

## Immunofluorescence for confocal imaging to measure MFI

Intact MM cells were stained with primary antibodies as described above. After PBS washes, cells were incubated with relevant fluorophore-conjugated secondary antibodies diluted in the same buffer for 30 min in the dark at room temperature. Cells were subsequently washed and resuspended in PBS, and incubated over coverslips (C9802; Sigma-Aldrich) for 30 min in the dark at room temperature. PBS was aspirated, and cells were incubated in 4% PFA in PBS for 15 min. Cells fixed to coverslips were washed with PBS and incubated in blocking buffer for 30 min at room temperature.

Cells are washed in PBS and incubated with DAPI (Thermo Fisher Scientific) resuspended in PBS to stain nuclei. Coverslips were mounted on glass slides using homemade glycerol-based mounting media.

## Image acquisition

Nikon Ti-2 inverted microscope equipped with a Yokogawa SoRa CSU-W1 super-resolution spinning disc confocal system, a Hamamatsu Flash V2 CMOS camera, and a Plan Apo 100× oil objective (NA = 1.45) was used for imaging. Z-sections were imaged every 0.2 $\mu$m. For super-resolution microscopy, the SoRa super-resolution mode with the same camera and objective was used. For MFI analysis, cells were imaged on a Nikon FN1 Upright Confocal microscope using a 60× oil immersion objective, and Z-stacks of up to thirteen 0.5-$\mu$m slices were captured and projected using the Fiji software.

## Image analysis

A Z-stack image containing only a single cell was used for image analysis. The number of foci of CH60 and TLR4 was counted using 3D Objects Counter in ImageJ with the same threshold setting for signal intensity and particle size for each image in the same experiment. The output images from 3D Objects Counter would show each single particle detected by the program, which can be used to determine the colocalized spots using Image Calculator in ImageJ. Line scans were performed with the Nikon Elements (NIS) software. 3D remodeling and a movie were made by the Imaris software (Bitplane). All plots and statistical analysis were performed by the Prism 9 software (GraphPad).

## Streptavidin-agarose coprecipitation assay

For biotinylated CH60 coprecipitation experiments, membrane fractions of MM cells prepared from biotin transfer assays as described above were used. From each experimental condition, equal protein amounts were aliquoted from fractions in which biotinylated CH60 was detected. The streptavidin resin and relevant membrane fractions were prepared as described above. An additional step to block the resin with 1% BSA in binding buffer (150 mM NaCl, 20 mM Tris–HCl, and 1 mM EDTA) was carried out. Subsequently, the blocked resin was incubated with the relevant membrane fraction and mixed end-to-end overnight at 4°C. Resin was then washed, boiled in Laemmli sample buffer, electrophoresed, and immunoblotted for biotinylated CH60 and its specific interacting partners.

## Lentiviral-mediated knockdown of CH60

HEK293T cells were plated at 40% confluency on 0.2% gelatin (G9391; Sigma-Aldrich)-coated cell culture dishes and incubated overnight. Media were then replaced with culturing media devoid of antibiotics. Lipofectamine transfection reagent (18324012; Invitrogen) was incubated in Opti-MEM (31985062; Gibco) at room temperature for 5 min. In an equal volume of Opti-MEM, optimized amounts of pCMV-dR8.91 (PVT2323; Life Science Market), pCMV-VSV-G (8454; Addgene), and CH60 shRNA (SHCLNG-NM_002156; MilliporeSigma) plasmids were mixed thoroughly. These reactions were then mixed together

and incubated at room temperature for 30 min. DNA–Lipofectamine complexes formed were pipetted onto the prepared HEK293T cells and incubated overnight. Media were then replaced with fresh antibiotic-free media for 48–72 h. Lentiviral particles in media produced by HEK293T cells were collected. Optimized amounts of protamine sulfate (P3369; Sigma-Aldrich) and lentiviral particles-containing media were mixed and pipetted onto MM cells in a six-well cell culture plate (140675; Thermo Fisher Scientific). MM cells were then "spinfected" at 1,000$g$ for 1 h at room temperature. Transduced MM cells were selected in fresh media supplemented with puromycin (P9620; Sigma-Aldrich), and knockdown efficiency was measured through immunoblotting as described above.

## siRNA-mediated knockdown of CH60

Hsp60 (HSPD1) human siRNA oligo duplex kit (SR302265; OriGene) containing three separate siRNA constructs and siTRAN transfection reagent (TT320002; OriGene) were used to transfect MM cells as per manufacturer's instructions. Universal scrambled negative control siRNA duplex (SR30004; OriGene) was transfected into MM cells in the same manner. Knockdown efficiency was measured through immunoblotting as described above.

## CRISPR-mediated mutant cell pool generation

Synthego's CRISPR design tool was used to design specific guide RNAs (gRNAs) that target an early exon and a common transcript. Chemically modified single-guide RNA (sgRNA) sequences were complexed together with the SpCas9 to form RNP and delivered into target cell lines via electroporation to obtain stable change in genome. Positive control sgRNA (RELA) was always transfected at the same time. The edited site is PCR-amplified, and amplicons are Sanger-sequenced. The sequence data were analyzed using Synthego's ICE software tool. ICE identifies the insertion–deletion (indel) frequency (ICE score) and the specific indels present in the pool. sgRNA and primer sequences used for respective target genes are as follows: TLR4, sgRNA: GCCUCAGGGGAUUAAAGCUC, Forward: TGCTGAGCACGTAGTAGGTG, Reverse: TGAACACCTCACCTTGTGCA; CD180, sgRNA: UUGUGUUUGGUAGAGUGUCA, Forward: TGTTTCTCCCATCAGAAAGAAGCCA, Reverse: ATTGATGCAACTGGTTTGAGAGC; TRAF6, sgRNA: GAAGCAGUGCAAACGCCAUG, Forward: GTCACTCCCAGTCTGCATT, Reverse: GGAACCTCTCCAGCTCATTT; and MyD88, sgRNA: GUCGGCCUACAGAGGCGCCA, Forward: CTAGCACCATCACCAGACCC, Reverse: CGCCTATCCGGACCTTTCTC.

## qRT–PCR analysis

Total RNAs from cells were purified by a NucleoSpin RNA II column (740955; Clontech) according to manufacturer's instruction. cDNAs were synthesized from the total RNAs using an iScript cDNA synthesis kit (1708891; Bio-Rad). qRT–PCR was performed and analyzed using a Bio-Rad CFX Connect real-time system. Relative expression was determined by $\Delta\Delta$Ct calculation and normalized to GAPDH mRNA levels of the same sample. The qRT–PCR primer sequences used for respective target transcripts are as follows: GAPDH, Forward: GTCTCCTCTGACTTCAACAGCG, Reverse: ACCACCCTGTTGCTGTAGCCAA; TLR-4, Forward: CCTGAGCTTTAATCCCCTGAGGC, Reverse:

CAGAAAAGGCTCCCAGGGCTA; BCL2, Forward: ATCGCCCTGTGGATGACT-GAGT, Reverse: GCCAGGAGAAATCAAACAGAGGC; BCL2A1, Forward: GGATAAGGCAAAACGGAGGCTG, Reverse: CAGTATTGCTTCAGGAGAGATAGC; cIAP2, Forward: GCTTTTGCTGTGATGGTGGACTC, Reverse: CTTGACG-GATGAACTCCTGTCC; IL-8, Forward: GAGAGTGATTGAGAGTGGACCAC, Reverse: CACAACCCTCTGCACCCAGTTT; IL-10, Forward: TCTCCGA-GATGCCTTCAGCAGA, Reverse: TCAGACAAGGCTTGGCAACCCA; NFKB1, Forward: GCAGCACTACTTCTTGACCACC, Reverse: TCTGCTCCTGAGCATTGACGTC; and NFKBIA, Forward: TCCACTCCATCCTGAAGGCTAC, Reverse: CAAGGACACCAAAAGCTCCACG.

## Trypan blue viability staining

$5 \times 10^5$ cells per well were plated in triplicate in a 24-well plate (142475; Thermo Fisher Scientific) and treated with appropriate combinations of drugs and recombinant fusion peptides. Post-treatment, the cells were incubated in HyClone Trypan Blue (SV30084.01; GE Healthcare), and viable cells were counted under a bright-field microscope. The assays were run in three biological replicates. The data were normalized to the mean of MBP-only control.

## In vivo mouse xenograft tumor model

Indicated MM cells were transplanted into the flanks of NOD/SCID/IL2rγnull (NSG) mice (005557; Jackson Laboratories) and grown to an initial volume of ~200 $mm^3$. On reaching baseline volume, 100 $\mu l$ of MBP-only control (425 ng/tumor) or MBP-PTR1 (500 ng/tumor) recombinant peptide in PBS was injected intratumorally a day before each intraperitoneal injection of bortezomib (1 mg/kg, days 0, 4, 7, 11, and 14). Tumor response was measured as mean tumor volume normalized to baseline tumor volume for each tumor. Some MBP- and MBP-PTR1–injected MM tumors were collected at 14 or 20 d and imaged for their gross morphology. Samples at day 14 were later sectioned and stained with H&E for further histological analysis. The experimental protocol for animal usage was reviewed and approved by the University of Wisconsin-Madison Institutional Animal Care and Use Committee, and all animal experiments were conducted in accordance with the University of Wisconsin-Madison Institutional Animal Care and Use Committee guidelines under the approved protocol. All mice were housed in specific pathogen-free conditions at the University of Wisconsin-Madison.

# Supplementary Information

# Acknowledgements

We thank the members of the Miyamoto laboratory for helpful comments on the project and the article. We thank Dr. Richard Burgess for the advice on protein purification, Dr. Alan Rapraeger for initial advice on the biotin transfer assay, and Dr. Caroline Alexander for helpful insights on the confocal images. This work was funded by NIH/NCI R01 CA155192 and R01 CA251595 (to S Miyamoto), a RIDE Scholar Award (to S Miyamoto), and a pilot fund from the UW Carbone Cancer Center Grant P30 CA014520 (to S Miya-moto). The Orbitrap instruments were purchased through the support of an NIH Shared Instrument Grant (NIH-NCRR S10RR029531 to L Li) and the University of Wisconsin-Madison, Office of the Vice Chancellor for Research and Graduate Education, with funding from the Wisconsin Alumni Research Foundation. L Li would like to acknowledge NIH grants RF1AG052324 and R01 DK071801. A Suzuki would like to acknowledge NIH grant R35GM147525. We also thank UW Surgery Histology Core for performing H&E staining of tumors and UW Optical Imaging Core (1S10OD025040-01) for use of the Nikon FN1 Upright Confocal microscope.

## Author Contributions

D De Bakshi: conceptualization, data curation, formal analysis, supervision, validation, investigation, visualization, methodology, project administration, and writing—original draft, review, and editing.
Y-C Chen: data curation, formal analysis, validation, investigation, visualization, methodology, and writing—review and editing.
SM Wuerzberger-Davis: data curation, formal analysis, validation, investigation, visualization, methodology, and writing—original draft, review, and editing.
M Ma: data curation, formal analysis, validation, investigation, visualization, methodology, and writing—original draft, review, and editing.
BJ Waters: data curation, validation, investigation, visualization, methodology, and writing—original draft, review, and editing.
L Li: resources, funding acquisition, methodology, project administration, and writing—original draft, review, and editing.
A Suzuki: resources, formal analysis, funding acquisition, visualization, methodology, project administration, and writing—review and editing.
S Miyamoto: conceptualization, resources, supervision, methodology, project administration, and writing—original draft, review, and editing.

## Conflict of Interest Statement

The authors declare that they have no conflict of interest.

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
