## [Reviewer comments · Life Science Alliance]

Life Science Alliance

Ectopic CH60 mediates HAPLN1-induced cell survival signaling in multiple myeloma

Debayan De Bakshi, Yu-Chia Chen, Shelly Wuerzberger-Davis, Min Ma, Bayley Waters, Lingjun Li, Aussie Suzuki, and Shigeki Miyamoto

DOI: <https://doi.org/10.26508/lsa.202201636>

Corresponding author(s): *Shigeki Miyamoto, University of Wisconsin-Madison*

Review Timeline:

Submission Date:	2022-07-27
Editorial Decision:	2022-09-02
Revision Received:	2022-11-04
Editorial Decision:	2022-11-29
Revision Received:	2022-11-29
Accepted:	2022-12-01

Transaction Report:

September 2, 2022

Re: Life Science Alliance manuscript #LSA-2022-01636-T

Prof. Shigeki Miyamoto
Wisconsin, University of
Dept. of Oncology
6159 WIMR
1111 Highland Avenue
Madison, Wisconsin 53705

Dear Dr. Miyamoto,

Thank you for submitting your manuscript entitled "HAPLN1 signals via a CH60-TLR4 complex to induce NF- κ B activity and drug resistance in myeloma cells" to Life Science Alliance. The manuscript was assessed by expert reviewers, whose comments are appended to this letter. We invite you to submit a revised manuscript addressing the Reviewer comments.

Thank you for this interesting contribution to Life Science Alliance. We are looking forward to receiving your revised manuscript.

Sincerely,

B. MANUSCRIPT ORGANIZATION AND FORMATTING:

Reviewer #1 (Comments to the Authors (Required)):

Roles of TLRs, and TLR4 in particular, in mediating the inflammatory response to endogenous "damage-associated molecular patterns" is well-documented. Study of De Bakshi et al. provides support for hypothesis that PTR-1, a structural domain of extracellular matrix protein HAPLN1, induces inflammatory response and cancer drug resistance, acting as a TLR4-activating DAMP. Authors develop (and take advantage of) an improved protocol for isolation of the PTR1 domain in a monomeric, more soluble, MBP-fused form. Biotin labeling demonstrated that PTR-1 binds a 55 kD plasma membrane protein, identified as chaperonin 60 by MS. Authors further propose that PTR1 activates NF- κ B through TLR4. This proposal is based on the following lines of evidence: CH60 and TLR4 coprecipitate; the PTR1-induced NF- κ B activation is susceptible to LPS-Rs, a known TLR4 inhibitor, and to the expression of dominant-negative TLR4, MyD88 and TRAF6 mutants. Overall, the manuscript suggests a new molecular mechanism for the DAMP-induced TLR4 activation. This mechanism may play a role in development of drug resistance in some forms of cancer. I think following question/concerns should be addressed before publication:

Does MD2 play any role in the PTR1/CH60/TLR4 axis?

CH60 was previously shown being able to signal through TLR4 independently of PTR1. In this light, what role PTR1 may play in the CH60-mediated TLR4 activation?

Will PTR1 treatment change the observed CH60 immunofluorescence pattern?

Dose, not stock concentration, of MBP proteins used in animal experiments should be given in figure legends.

In light of reference that other 'matrikines' "are capable of engaging a wide variety of receptors..." demonstration of capability to inhibit PTR-1-induced responses with an inhibitor of downstream TLR4 signaling (not ones that act at the level of receptor-ligand interactions) would strengthen manuscript's conclusions.

Discussion is not well structured, excessively long, and requires additional editing.

Reviewer #2 (Comments to the Authors (Required)):

This study by Bakshi et al is an extension of their previous study showing that the PTR1 domain of HAPLN1 activates NF- κ B signaling and contributes to drug resistance to bortezomib in multiple myeloma cells (Huynh et al, 2018). In the present study, the authors identified the CH60-TLR4 complex as the plasma membrane receptor of HAPLN1-PTR1. Specifically, the paper proposes that HAPLN1-PTR1 binds to CH60, also known as HSPD1, a chaperon protein normally expressed in mitochondria that is aberrantly localized on the plasma membrane. Upon binding, this interaction subsequently activates NF- κ B signaling and up-regulates the expression of anti-apoptotic genes via TLR4/MyD88/TRAF6. The manuscript flows well. However, the study suffers from several critical limitations in the experimental design, which impair the reliability of the conclusions.

Major

- The authors used a bacterial system for purifying GST-fused PTR1 peptide. Although poly-L-lysine was used to remove endotoxin and Figure 1E shows the ability of MBP-PTR1 to activate NF- κ B in the presence of lipid A inhibitor, possible endotoxin contamination during the purification process remains a major concern. This should be overcome by performing a confirmatory experiment in which purification using a mammalian system is used in order to generate recombinant PTR1 (for instance, by fusing PTR1 with a secretion signal peptide).

- A major finding of this study is CH60 is the plasma membrane receptor of HAPLN1-PTR1. However, the data of CH60 localization is not convincing. The patchy distribution pattern on the plasma membrane is not typical for a plasma membrane protein. Although the authors explained that this might be due to the fluidity of the cell membrane, it seems incongruent with the localization of other known plasma membrane proteins. Given what is known about HSPD1, we have great skepticism about this claim and more robust confirmation is absolutely required. First, with regards to the staining experiments, the use of an isotype

only control excludes the possibility that the staining is due to background from the secondary antibody but does not exclude the possibility of non-specific binding of primary antibody to off-target proteins. To address this, it is crucial to develop CH60-knockout cells (not just the 50% knockdown cells that were used in the current version), and show that the staining seen is lost in these cells. Rescue experiment by re-introducing a tagged version of CH60 into the knockout cells, will help confirm the localization and function of CH60. Moreover, super-resolution confocal microscopy or electron microscopy are needed to better resolve the images and make sure that the dot distribution is truly in the membrane itself and not below the membrane (like in mitochondria close to the cell surface, for example). Of note, it is possible that CH60 is involved in the folding of TLR4 for the localization of TLR4 on the plasma membrane, and therefore CH60 does not necessarily have to be expressed on the plasma membrane for TLR4 signaling to be affected. Therefore, the expression and localization of TLR4 should also be tested in the CH60 knockout cells.

- EMSA was used as the only method to determine NF- κ B activation in this study. EMSA shows NF- κ B nuclear localization and binding to DNA with a κ B sequence. However, it does not fully reflect all steps in NF- κ B activation: IKK activation, I κ B phosphorylation and degradation, NF- κ B translocation from cytoplasm to nucleus, the detailed subunit composition of the NF- κ B complexes in the nucleus, and the subsequent transcriptional activation of NF- κ B-regulated genes. Therefore, EMSA should not be used in isolation. Other readouts should be used to confirm the results. In this case, at minimum, I κ B phosphorylation and degradation, as well as mRNA induction for NF- κ B regulated genes must be shown.

- Similarly, to confirm the role of TLR4, MyD88, and TRAF6 in HAPLN1-PTR1-induced NF- κ B activation, the authors should use complete knockout cells followed by rescue experiment, rather than mixed clones. This should be achievable considering that they have targeting gRNAs that can reduce gene expression.

- Supplemental Figure 1 shows a notable amount of NF- κ B signals sensitive to RelB antibody at baseline in MM1R and U266 cells, indicating active non-canonical NF- κ B signaling in these cells. The exact experimental procedure could not be found in the paper for us to fully understand what was done, but we surmised that this was a RelB IP followed by EMSA using the left over material (perhaps it was a supershift that was narrowly cropped - can't tell). This is not mentioned or discussed in the paper, which was confusing. Second, noncanonical NF- κ B activation is not an expected outcome of TLR4 signaling, so that was confusing as well. The authors need to clarify this part of the paper, including better description of the methods, and if they find this is relevant, they should examine if there is any contribution of non-canonical signaling in RPMI8226 cells, which is used as the primary model in the current study.

Minor

- When presenting EMSA results, at least once in the paper, the authors should show the entire gel that includes the free probes, NF- κ B shifted, and non-specific bands (if any), rather than the NF- κ B shifts alone. This would provide confidence on the methodological aspects of performing an EMSA. The multiple additional iterations shown in the paper can then be cropped regions of the gels.

- Figure 2G (imaging of flank myeloma tumors in mice): Even if taking into consideration the degree of tissue necrosis, the magnification of MBP+BORT and MBP-PTR1+BORT do not appear to be the same. Please verify the magnification.

- Figure 3C and 3D (post-biotinylation material resolved by SDS-PAGE and streptavidin HRP): These panels should be labelled with more details in the figures themselves so that the reader can tell the difference between these panels which currently appear to be depicting the same materials, when in fact 3C is the material after gel filtration and 3D is the material after Streptavidin-agarose pull down.

- Figure 6A (gene expression of anti-apoptotic genes after treatment with PTR1): TLR4-mt (CRISPR/Cas9 pool of cells with TLR4 deletion) alone may be sufficient to change the expression of anti-apoptotic genes, even prior to PTR1 stimulation. The relative expression level of these genes should also be shown in TLR4-mt vs control prior to PTR1. Ideally, this panel should include four groups for each gene (WT +/- MBP-PTR1 and TLR4-mt +/- MBP-PTR1).

Reviewer #3 (Comments to the Authors (Required)):

The paper from De Bakshi et al provides compelling molecular and cellular data to indicate that HAPLN1, previously studied as an activator of NF-kappaB and bortezomib resistance in multiple myeloma that is expressed in bone marrow stroma, engages a cell surface protein (CH60) to elicit its effects. CH60 is convincingly shown to interact with TLR4 to drive NF-kappaB activation. The work is an important extension of the previous 2018 study. The reviewer commends the authors on a nicely written and well-controlled study. I have a few minor comments and observations.

1) The main text should indicate that the NF-kappaB assay is an EMSA, as this assay is not used to the extent that it was used years ago. The authors are commended on the use of this assay.

2) Fig EV1 shows the use of a RelB antibody to block the NF-kappaB EMSA complex which seems to eliminate the major shifted complex. Does this result indicate that the TLR4 response is largely a non-canonical NF-kappaB signal? Are canonical signals not activated? Is there a molecular mark/assay result to show the non-canonical activation response by PTR1?

3) For the cell lines that didn't show a strong response to PTR1 (p. 6) are they consistently low in CH60 expression?

4) In the bortezomib study in Fig. 2 - does bortezomib block NF-kappaB activation? Shouldn't it block the PTR1 response, or is this simply a dose competition?

5) Can HAPLN1 (or rather PTR1) be visualized as co-localizing with CH60 and TLR4 at the cell surface? Does PTR1 binding to

CH60 enhance its interaction with TLR4?

We thank the reviewers for their thoughtful comments. We believe that these suggestions have greatly improved the quality of our manuscript. We have now addressed all comments from the reviewers and outline these changes point by point below.

Reviewer #1:

Roles of TLRs, and TLR4 in particular, in mediating the inflammatory response to endogenous "damage-associated molecular patterns" is well-documented. Study of De Bakshi et al. provides support for hypothesis that PTR-1, a structural domain of extracellular matrix protein HAPLN1, induces inflammatory response and cancer drug resistance, acting as a TLR4-activating DAMP. Authors develop (and take advantage of) an improved protocol for isolation of the PTR1 domain in a monomeric, more soluble, MBP-fused form. Biotin labeling demonstrated that PTR-1 binds a 55 kD plasma membrane protein, identified as chaperonin 60 by MS. Authors further propose that PTR1 activates NF- κ B through TLR4. This proposal is based on the following lines of evidence: CH60 and TLR4 coprecipitate; the PTR1-induced NF- κ B activation is susceptible to LPS-Rs, a known TLR4 inhibitor, and to the expression of dominant-negative TLR4, MyD88 and TRAF6 mutants. Overall, the manuscript suggests a new molecular mechanism for the DAMP-induced TLR4 activation. This mechanism may play a role in development of drug resistance in some forms of cancer. I think following question/concerns should be addressed before publication:

Comment 1: Does MD2 play any role in the PTR1/CH60/TLR4 axis?

Response 1: *We have addressed this query by performing a co-precipitation experiment to evaluate whether MD-2 interacts with components of the PTR1/CH60/TLR4 axis, and by chemical inhibiting MD-2 to evaluate its role in PTR1-induced NF- κ B activity. In new Fig. S5A, we show immunoblot analysis with anti-MD-2 antibody of streptavidin-agarose precipitated membrane fractions post gel filtration, as in Fig. 7A. The data now show that streptavidin-agarose pull down of PTR1-induced cell surface biotinylated fraction contains CH60, TLR4 and MD-2, but not other surface proteins such as CD14 and TLR9 (new Fig. 7A and new S5A). These results suggest that CH60 specifically interact with TLR4 and MD-2 in the plasma membrane fraction of multiple myeloma (MM) cells. Moreover, pre-treatment of RPMI8226 MM cells with MD2-in-1, a MD2-specific chemical inhibitor (Zhang et al., 2016. PMID: 27118147), abrogated MBP-PTR1-induced NF- κ B activity while leaving TNF α -induced NF- κ B activity intact (new Fig. 7D). Taken together, these new results supported the concept that MD-2 can modulate NF- κ B activity stimulated by the PTR1/CH60/TLR4 axis.*

Comment 2: CH60 was previously shown being able to signal through TLR4 independently of PTR1. In this light, what role PTR1 may play in the CH60-mediated TLR4 activation?

Response 2: *We acknowledge and cite previous studies (Ohashi et al., 2000. PMID: 10623794; Cohen-Sfady et al., 2005. PMID: 16148103) that show that exogenously-supplied recombinant CH60 in the culture media can activate TLR4. Exogenously added CH60 at some arbitrary amounts may or may not reflect a true physiological signaling mechanism. Moreover, none of these publications provided any evidence that TLR4 and CH60 physically interact to signal. In response to reviewer #2 (see below), we performed additional super-resolution imaging experiments, and are the first to show that that endogenous CH60 is physically associated with TLR4 in pre-formed complexes at the MM plasma membrane (new Fig. 6A). Additionally, knockdown of endogenous CH60 in MM cells significantly reduced levels of cell surface CH60 (new Fig. S3C and D) but failed to alter basal, unstimulated levels of NF- κ B activity in MM cells (Fig. 5C). This suggests that endogenous CH60 complexed with TLR4 does not independently activate NF- κ B. Our manuscript differentiates between exogenously supplied*

recombinant CH60 which may activate the TLR4 signaling cascade and endogenously expressed CH60 identified at the MM plasma membrane that shows no apparent basal stimulatory properties of the TLR4 signaling cascade but represents a HAPLN1-PTR1 receptor.

Comment 3: Will PTR1 treatment change the observed CH60 immunofluorescence pattern?

Response 3: *We performed additional immunofluorescence and super-resolution imaging experiments to answer the query. PTR1 stimulation of MM cells does not change the CH60 pattern in terms of either the number of CH60 puncta present or the frequency of colocalization between CH60 and TLR4 at the MM cell surface (new Fig. 6B-6D). Thus, our data support the idea that PTR1 activates a pre-assembled CH60-TLR4 complex to signal downstream.*

Comment 4: Dose, not stock concentration, of MBP proteins used in animal experiments should be given in figure legends.

Response 4: *We now include the doses of both MBP (control) and MBP-PTR1 injected into individual tumors, and not stock concentration, are now included in the corresponding figure legends and materials and methods sections.*

Comment 5: In light of reference that other 'matrikines' "are capable of engaging a wide variety of receptors..." demonstration of capability to inhibit PTR1-induced responses with an inhibitor of downstream TLR4 signaling (not ones that act at the level of receptor-ligand interactions) would strengthen manuscript's conclusions.

Response 5: *We agree with the reviewer's evaluation of this, and have performed additional experiments to interrogate the specificity of the PTR1 matrikine engaging TLR4 signaling. TAK-242, a well-described inhibitor of TLR4 signaling that blocks interaction of TLR4 with downstream adaptor molecules such as MyD88 (Matsunaga et al., 2011. PMID:20881006), was shown to block MBP-PTR1-induced NF- κ B activity while leaving TNF α -induced NF- κ B activity intact (new Fig. 7E). To further interrogate well-defined downstream mediators of TLR4 signaling, CRISPR-generated mutant MyD88 and TRAF6 RPMI8226 cells were shown to be incapable of mounting an NF- κ B response upon PTR1 stimulation, while leaving TNF α -mediated NF- κ B responses intact. Collectively, chemical inhibitor and genetic mutant analyses indicate that cell surface CH60 directly recognizes HAPLN1-PTR1 and signals via a TLR4/MyD88/TRAF6 cascade to cause NF- κ B activation in MM cells.*

Comment 6: Discussion is not well structured, excessively long, and requires additional editing.

Response 6: *We now present a more concise version of the discussion section to make the points made more clearer.*

Reviewer #2:

This study by Bakshi et al is an extension of their previous study showing that the PTR1 domain of HAPLN1 activates NF- κ B signaling and contributes to drug resistance to bortezomib in multiple myeloma cells (Huynh et al, 2018). In the present study, the authors identified the CH60-TLR4 complex as the plasma membrane receptor of HAPLN1-PTR1. Specifically, the paper proposes that HAPLN1-PTR1 binds to CH60, also known as HSPD1, a chaperon protein normally expressed in mitochondria that is aberrantly localized on the plasma membrane. Upon binding, this interaction subsequently activates NF- κ B signaling and up-regulates the expression of antiapoptotic genes via TLR4/MyD88/TRAF6. The

manuscript flows well. However, the study suffers from several critical limitations in the experimental design, which impair the reliability of the conclusions.

Major

Comment 1: The authors used a bacterial system for purifying GST-fused PTR1 peptide. Although poly-L-lysine was used to remove endotoxin and Figure 1E shows the ability of MBP-PTR1 to activate NF- κ B in the presence of lipid A inhibitor, possible endotoxin contamination during the purification process remains a major concern. This should be overcome by performing a confirmatory experiment in which purification using a mammalian system is used in order to generate recombinant PTR1 (for instance, by fusing PTR1 with a secretion signal peptide).

Response 1: *We share the concern for endotoxin contamination in our recombinant protein preparations. In our earlier publication (Huynh et al., 2018), we leveraged the use of a pSecTag2a vector, which contains a secretion signal peptide, and transfected HAPLN1 into host HEK293 cells to show that HAPLN1 produced from a mammalian source was able to activate NF- κ B in MM cells (in fact, this was how HAPLN1 was originally discovered to signal NF- κ B in MM cells). We attempted to produce the PTR1 domain of HAPLN1 using the same pSecTag2a vector using a similar approach, but the PTR1 domain was not secreted and remained sequestered in the cytoplasm (likely due to inappropriate glycosylation events). To overcome this technical limitation to interrogate each HAPLN1 domain, we used a bacterial system to generate recombinant peptides. In addition to the poly-L-lysine endotoxin depletion, as noted by reviewer #2, we used polymyxin-B, a known inhibitor of endotoxin, to confirm that MBP-PTR1-induced NF- κ B activity was not coming from endotoxin contamination, but from the peptide itself (Fig. 1E). Furthermore, to rule out the possibility that NF- κ B activation was due to the presence of contaminating endotoxin, we measured the amounts of endotoxin in MBP-PTR1 preparations (new Fig. S1A). The levels of contaminating endotoxin were found to be <0.15 ng/mL, and the final endotoxin concentration when RPMI8226 cells were treated was ~ 4 orders of magnitude below the threshold required for detectable NF- κ B activation in RPMI8226 cells as we previously reported (Huynh et al., 2018). This confirms that HAPLN1-PTR1 is responsible for the NF- κ B activity stimulated by MBP-PTR1 preparations, and not endotoxin contamination nor the MBP-fusion tag.*

Comment 2: A major finding of this study is CH60 is the plasma membrane receptor of HAPLN1-PTR1. However, the data of CH60 localization is not convincing. The patchy distribution pattern on the plasma membrane is not typical for a plasma membrane protein. Although the authors explained that this might be due to the fluidity of the cell membrane, it seems incongruent with the localization of other known plasma membrane proteins. Given what is known about HSPD1, we have great skepticism about this claim and more robust confirmation is absolutely required. First, with regards to the staining experiments, the use of an isotype only control excludes the possibility that the staining is due to background from the secondary antibody but does not exclude the possibility of non-specific binding of primary antibody to off-target proteins. To address this, it is crucial to develop CH60-knockout cells (not just the 50% knockdown cells that were used in the current version), and show that the staining seen is lost in these cells. Rescue experiment by reintroducing a tagged version of CH60 into the knockout cells, will help confirm the localization and function of CH60. Moreover, super-resolution confocal microscopy or electron microscopy are needed to better resolve the images and make sure that the dot distribution is truly in the membrane itself and not below the membrane (like in mitochondria close to the cell surface, for example). Of note, it is possible that CH60 is involved in the folding of TLR4 for the localization of TLR4 on the plasma membrane, and therefore CH60 does not necessarily have to be expressed on the plasma membrane for TLR4 signaling to be affected. Therefore, the expression and localization of TLR4 should also be tested in the CH60 knockout cells.

Response 2: *We thank the reviewer for suggestions to reinforce the finding that CH60 is detected on the plasma membrane on of MM cells. Since CH60 is a common essential gene (Christensen et al., 2010. PMID: 20393889; Perezgasga et al, 1999. PMID: 10456322), a complete knockout of CH60 would be lethal. Consistent with these reports, we titrated a CH60 knock down to ~50% of control wildtype levels (Fig. 5A) as lower expression made MM cells unviable. To address the issue of antibody specificity, we performed immunofluorescence in intact control shRNA and CH60 shRNA RPMI8226 cells. Mean fluorescence intensity (MFI) analysis showed that anti-CH60 antibody staining generated a significantly weaker signal in CH60 knockdown cells compared to control cells (new Fig. S3D). This reduction in cell surface signal in RPMI 8226 cells was consistent with reduction in biotinylation of cell surface CH60 in the MBP-PTR1 biotinylation assay (Fig. 5B). These results collectively suggest that CH60 levels on the cell surface are indeed decreasing in CH60 knockdown cells, and that antibody is specifically staining CH60 on the plasma membrane.*

As suggested by reviewer #2, we now performed super-resolution microscopy and subsequent intensity profile analysis. Super-resolution imaging detected cell surface CH60 on MM cells as “patchy” puncta (new Fig. 4A) studded through the MM plasma membrane which was uniformly stained using monoclonal anti-CD138 antibody, a bona-fide positive control for cell surface localization in MM cells. An image of a mitotic cell was included to clearly depict the unstained cytoplasmic region in these unpermeablized MM cells. Furthermore, intensity profile plots of CH60 and CD138 signals generated using super-resolution imaging (new Fig. 4B) showed clear visualization of the colocalization of these two proteins, suggesting that CH60 is aberrantly localized at the MM plasma membrane. Images of 5 consecutive sections along the Z-plane of the same cells as in new Fig. 4A reveal that multiple CH60 puncta are present in distinct locations at different planar regions of the plasma membrane (new Fig. 4C). A 3D movie rendered using the different planes of an intact MM cell captured by super-resolution microscopy (new Video 1) confirm that the CH60 puncta colocalizes with CD138 staining, and exists studded through the plasma membrane.

We also performed colocalization studies to assess the relative expression and positionality of CH60 to TLR4 on the cell surface. Confocal images of different Z-planes showed that a subset of cell surface CH60 and TLR4 colocalized with each other in puncta-like complexes on the MM cell surface, and intensity profile plots of the CH60 and TLR4 signals displayed clear visualization of the colocalization of these two proteins (new Fig. 6A). Additionally, about ~ 65% of the CH60 puncta and ~ 60% of TLR4 puncta colocalize with each other (new Fig. 6B and C). Interestingly, a proportion of both CH60 and TLR4 puncta was seen unassociated with each other on the plasma membrane, suggesting that they can also exist independently at the myeloma cell surface. The existence of a subpopulation of CH60-only and TLR4-only puncta indicates that physical association between CH60 and TLR4 is not required for either’s cell surface localization. The significant subpopulation of CH60-only and TLR4-only puncta also suggests that the antibodies do not non-specifically bind to a common off-target protein. As to the patchy puncta staining of CH60 on the cell surface, we initially had ascribed it to the fluidity of the cell membrane. However, the fact that TLR4 staining of unpermeablized, intact MM cells also displays a patchy punctate pattern, in contrast to the uniform staining pattern CD138 displays, indicates that CH60 and TLR4 exists in pre-formed foci-like complexes on the MM cell surface.

Comment 3: EMSA was used as the only method to determine NF- κ B activation in this study. EMSA shows NF- κ B nuclear localization and binding to DNA with a κ B sequence. However, it does not fully reflect all steps in NF- κ B activation: IKK activation, I κ B phosphorylation and degradation, NF- κ B translocation from cytoplasm to nucleus, the detailed subunit composition of the NF- κ B complexes in the nucleus, and the subsequent transcriptional activation of NF- κ B-regulated genes. Therefore, EMSA should not be used in isolation. Other readouts should be used to confirm the results. In this case, at minimum, I κ B phosphorylation and degradation, as well as mRNA induction for NF- κ B regulated genes must be shown.

Response 3: *To confirm that NF- κ B is activated by MBP-PTR1, in addition to EMSA analysis, we show using immunoblot that MBP-PTR1 also concordantly induced phosphorylation of the inhibitor of nuclear factor- κ B (I κ B) kinase (IKK) complex and degradation of I κ B α in RPMI8226 cells (new Fig. S1B). Simultaneously, pretreatment of RPMI8226 cells with IKK16, a chemical inhibitor of the I κ B kinase (IKK) complex (Waelchli et al., 2006. PMID: 16236504), was shown to block MBP-PTR1-mediated stimulation of the NF- κ B signaling cascade (new Fig. S1B). We also observed that MBP-PTR1 was able to induce transcription of NF- κ B-regulated target genes (new Fig. S1C), I κ B α (NFKBIA) (Sun et al., 1993. PMID: 8096091) and p105 (NFKB1) (Ten et al, 1992. PMID: 1740105). In our earlier publication, we have shown through super-shift assay that PTR1-induces a canonical NF- κ B complex comprising p65 and p50 (Huynh et al., 2018). These results collectively indicate that MBP-PTR1 possesses the ability to evoke canonical elements of the NF- κ B signaling cascade.*

Comment 4: Similarly, to confirm the role of TLR4, MyD88, and TRAF6 in HAPLN1-PTR1-induced NF- κ B activation, the authors should use complete knockout cells followed by rescue experiment, rather than mixed clones. This should be achievable considering that they have targeting gRNAs that can reduce gene expression.

Response 4: *We wanted to perform experiments suggested and thus tried to generate single clones from the mutant MM pools, but repeatedly faced a technical barrier. Unfortunately, MM cell lines, including RPMI8226 cells and MM.1R cells, are highly resistant to single clone isolation procedures. When isolated as a single cell, these suspension cells do not proliferate, and undergo senescence or cell death after a few divisions. Please refer to “response 5” made to reviewer 1, in which we used chemical inhibitors to bolster our manuscript’s conclusions.*

Comment 5: Supplemental Figure 1 shows a notable amount of NF- κ B signals sensitive to RelB antibody at baseline in MM1R and U266 cells, indicating active non-canonical NF- κ B signaling in these cells. The exact experimental procedure could not be found in the paper for us to fully understand what was done, but we surmised that this was a RelB IP followed by EMSA using the left over material (perhaps it was a supershift that was narrowly cropped - can't tell). This is not mentioned or discussed in the paper, which was confusing. Second, noncanonical NF- κ B activation is not an expected outcome of TLR4 signaling, so that was confusing as well. The authors need to clarify this part of the paper, including better description of the methods, and if they find this is relevant, they should examine if there is any contribution of non-canonical signaling in RPMI8226 cells, which is used as the primary model in the current study.

Response 5: *Reviewer #2 is correct that the EMSA in Supplemental Figure 1 is a super-shift with RelB antibody, and we agree that the description of this experiment should have been clearer. To that end, we include a wider view of the basal, unstimulated RelB complex being super-shifted in the relevant cell lines (Fig. S1D-F). Where indicated, EMSA was performed in the presence of RelB antibody to super-shift basal RelB complexes (which are not activated by HAPLN1-PTR1) to enable quantification of canonical NF- κ B complexes activated by HAPLN1-PTR1, as reported previously (Huynh et al., 2018). All EMSA are performed in the presence of excess probe to ensure that availability of probe is not a limiting factor in quantifying the relative activation of the transcription factor being measured (new Fig. 1B). We also include a more comprehensive description of the super-shift assay in the figure legend for Supplemental Figure 1, and in the Materials and Methods. We do not report any contribution of non-canonical NF- κ B activity downstream of TLR4 signaling in this manuscript.*

Minor

Comment 1: When presenting EMSA results, at least once in the paper, the authors should show the entire gel that includes the free probes, NF- κ B shifted, and non-specific bands (if any), rather than the

NF- κ B shifts alone. This would provide confidence on the methodological aspects of performing an EMSA. The multiple additional iterations shown in the paper can then be cropped regions of the gels.

Response 1: *As requested, we now include the entire gels for both NF- κ B and Oct-1 in the first EMSA images shown in the manuscript (new Fig. 1B), with the free probe line indicated. All EMSA are performed in the presence of excess probe to ensure that availability of probe is not a limiting factor in quantifying the relative activation of the transcription factors being measured. Multiple additional iterations of EMSA are shown with only the shifted transcription factor areas being depicted.*

Comment 2: Figure 2G (imaging of flank myeloma tumors in mice): Even if taking into consideration the degree of tissue necrosis, the magnification of MBP+BORT and MBP-PTR1+BORT do not appear to be the same. Please verify the magnification.

Response 2: *We can confirm the magnification is indeed the same in Fig. 2G.*

Comment 3: Figure 3C and 3D (post-biotinylation material resolved by SDS-PAGE and streptavidin HRP): These panels should be labelled with more details in the figures themselves so that the reader can tell the difference between these panels which currently appear to be depicting the same materials, when in fact 3C is the material after gel filtration and 3D is the material after Streptavidin-agarose pulldown.

Response 3: *We have provided more detailed labels in Figure 3 to improve clarity.*

Comment 4: Figure 3C and 3D (post-biotinylation material resolved by SDS-PAGE and streptavidin HRP): These panels should be labelled with more details in the figures themselves so that the reader can tell the difference between these panels which currently appear to be depicting the same materials, when in fact 3C is the material after gel filtration and 3D is the material after Streptavidin-agarose pulldown.

Response 4: *We have provided more detailed labels in Figure 3 to improve clarity.*

Comment 5: Figure 6A (gene expression of anti-apoptotic genes after treatment with PTR1): TLR4-mt (CRISPR/Cas9 pool of cells with TLR4 deletion) alone may be sufficient to change the expression of anti-apoptotic genes, even prior to PTR1 stimulation. The relative expression level of these genes should also be shown in TLR4-mt vs control prior to PTR1. Ideally, this panel should include four groups for each gene (WT +/- MBP-PTR1 and TLR4-mt +/- MBP-PTR1).

Response 5: *To address this, we performed additional analysis of the same data used to generate Fig. 6A to measure basal differences in mRNA levels of anti-apoptotic genes between wildtype (WT) and TLR4-mt cells, in the absence of MBP-PTR1 stimulation. Upon MBP-only (control) treatment, TLR4-mutant RPMI8226 cells showed negligible differences in the basal levels of these NF- κ B-regulated genes compared to their wildtype counterparts (new Fig. S5J), suggesting that TLR4 mutation does not basally cause any potent transcriptional changes in RPMI8226 cells.*

Reviewer #3:

The paper from De Bakshi et al provides compelling molecular and cellular data to indicate that HAPLN1, previously studied as an activator of NF-kappaB and bortezomib resistance in multiple myeloma that is expressed in bone marrow stroma, engages a cell surface protein (CH60) to elicit its effects. CH60 is convincingly shown to interact with TLR4 to drive NF-kappaB activation. The work is an important extension of the previous 2018 study. The reviewer commends the authors on a nicely written and well-controlled study. I have a few minor comments and observations.

Comment 1: The main text should indicate that the NF-kappaB assay is an EMSA, as this assay is not used to the extent that it was used years ago. The authors are commended on the use of this assay.

Response 1: *We have now included the following in the main text alongside Fig. 1B: “Experiments measuring relative changes in NF-κB activity are performed using Electrophoretic Mobility Shift Assay (EMSA) analysis, unless otherwise specified. The first EMSA included (Fig. 1B) displays the entire gel, showing the shifted transcription factors relative to the free probe line. All EMSA are performed in the presence of excess probe to ensure that availability of probe is not a limiting factor in quantifying the relative activation of the transcription factor being measured. Subsequent EMSA gels are cropped to include only the shifted transcription factors.”*

Comment 2: Fig EV1 shows the use of a RelB antibody to block the NF-kappaB EMSA complex which seems to eliminate the major shifted complex. Does this result indicate that the TLR4 response is largely a non-canonical NF-kappaB signal? Are canonical signals not activated? Is there a molecular mark/assay result to show the non-canonical activation response by PTR1?

Response 2: *We apologize for the lack of clarity regarding the original Supplementary Figure, as reviewer #2 also expressed a similar confusion about the EMSA assays performed in the presence of RelB antibody. The EMSAs being referred to by reviewers in Supplemental Figure 1 are super-shift assays with RelB antibody, and we agree that the description of this experiment should have been clearer. To that end, we include a wider view of the basal, unstimulated RelB complex being super-shifted in the relevant cell lines (Fig. S1D-F). Where indicated, EMSA was performed in the presence of RelB antibody to super-shift basal RelB complexes (which are not activated by HAPLN1-PTR1) to enable quantification of canonical NF-κB complexes activated by HAPLN1-PTR1, as reported previously (Huynh et al., 2018). All EMSA are performed in the presence of excess probe to ensure that availability of probe is not a limiting factor in quantifying the relative activation of the transcription factor being measured (Fig. 1B). We also include a more comprehensive description of the super-shift assay in the figure legend for Supplemental Figure 1, and in the Materials and Methods. We do not report any contribution of non-canonical NF-κB activity downstream of TLR4 signaling in any cell line in this manuscript.*

Comment 3: For the cell lines that didn't show a strong response to PTR1 (p. 6) are they consistently low in CH60 expression?

Response 3: *We performed an additional immunoblot, and note that cell lines that show a weaker response to PTR1 (Fig. S1D-F) compared to strongly-responsive RPMI8226 cells (Fig. 1F), indeed have relatively lower levels of overall CH60 expression compared to RPMI8226 cells (new Fig. S2D). In fact, PTR1-unresponsive U266 cells have the lowest CH60 expression amongst all MM cell lines tested.*

Comment 4: In the bortezomib study in Fig. 2 - does bortezomib block NF-kappaB activation? Shouldn't it block the PTR1 response, or is this simply a dose competition?

Response 4: *We appreciate the reviewer for this insightful question. In our earlier publication (Huynh et al., 2018), we showed that PTR1 was able to induce bortezomib-resistant NF-κB activity, and that this activity persisted despite bortezomib effectively inhibiting the catalytic activity of the 26S proteasome. In that study, we also showed that the presence of PTR1 did not reduce bortezomib efficacy or increase proteasome activity. In future studies, we aim to parse out the downstream mechanism in greater detail to identify the exact mechanism of bortezomib-inhibitor resistant NF-κB activity.*

Comment 5: Can HAPLN1 (or rather PTR1) be visualized as co-localizing with CH60 and TLR4 at the cell surface? Does PTR1 binding to CH60 enhance its interaction with TLR4?

Response 5: *To address if PTR1 alters CH60-TLR4 engagement at the cell surface, we initially interrogated whether CH60 basally forms a complex with TLR4 on the surface of MM cells. Live, unpermeabilized RPMI8226 cells were co-stained with anti-CH60 and anti-TLR4 antibodies for immunofluorescence analysis. Confocal images of different Z-planes showed that a subset of cell surface CH60 and TLR4 colocalized with each other in puncta-like complexes on the MM cell surface (new Fig. 6A). Furthermore, the intensity profile plots of the CH60 and TLR4 signals displayed clear visualization of the colocalization of these two proteins (new Fig. 6A). Additionally, stimulation of RPMI8226 cells with PTR1 did not alter the efficiency of colocalization between CH60 and TLR4, with ~ 65% of the CH60 puncta and ~ 60% of TLR4 puncta remaining colocalized before and after MBP-PTR1 treatment (new Fig. 6B and C). Interestingly, a proportion of both CH60 and TLR4 puncta was seen unassociated with each other on the plasma membrane, suggesting that they can also exist independently at the myeloma cell surface. Additionally, PTR1 stimulation did not alter the number of CH60 and TLR4 puncta present on the MM cell surface, with ~ 45 puncta of CH60 and ~ 50 puncta of TLR4 being present before and after PTR1 treatment (new Fig. 6B and D). Collectively, this suggests that a subset of CH60 and TLR4 exist as preformed complexes present on the MM cell surface whose colocalization and cell surface expression is independent of PTR1 stimulation.*

November 29, 2022

RE: Life Science Alliance Manuscript #LSA-2022-01636-TR

Prof. Shigeki Miyamoto
University of Wisconsin-Madison
Dept. of Oncology
6159 WIMR
1111 Highland Avenue
Madison, Wisconsin 53705

Dear Dr. Miyamoto,

Thank you for submitting your revised manuscript entitled "Ectopic CH60 mediates HAPLN1-induced cell survival signaling in multiple myeloma". We would be happy to publish your paper in Life Science Alliance pending final revisions necessary to meet our formatting guidelines.

- please add ORCID ID for secondary corresponding author-they should have received instructions on how to do so
- please make sure that the author order in our system and in the manuscript file match

Figure Check:

- please include sizes next to all blots

A. FINAL FILES:

B. MANUSCRIPT ORGANIZATION AND FORMATTING:

**Submission of a paper that does not conform to Life Science Alliance guidelines will delay the acceptance of your

manuscript.**

The license to publish form must be signed before your manuscript can be sent to production. A link to the electronic license to publish form will be sent to the corresponding author only. Please take a moment to check your funder requirements.

Sincerely,

Reviewer #1 (Comments to the Authors (Required)):

Authors satisfactory responded to previous critiques and present an improved manuscript suitable for publication. Some pages of Supplemental materials are scrawled and should be reformatted.

Reviewer #3 (Comments to the Authors (Required)):

The authors have successfully addressed my concerns. I recommend publication.

December 1, 2022

RE: Life Science Alliance Manuscript #LSA-2022-01636-TRR

Prof. Shigeki Miyamoto
University of Wisconsin-Madison
Dept. of Oncology
6159 WIMR
1111 Highland Avenue
Madison, Wisconsin 53705

Dear Dr. Miyamoto,

Thank you for submitting your Research Article entitled "Ectopic CH60 mediates HAPLN1-induced cell survival signaling in multiple myeloma". It is a pleasure to let you know that your manuscript is now accepted for publication in Life Science Alliance. Congratulations on this interesting work.

DISTRIBUTION OF MATERIALS:

Again, congratulations on a very nice paper. I hope you found the review process to be constructive and are pleased with how the manuscript was handled editorially. We look forward to future exciting submissions from your lab.

Sincerely,
